ecology/environmental science

soundscape, bioacoustics, richness, ecosystem health

**Author for correspondence:**
T. Aran Mooney
e-mail: amooney@whoi.edu

# Listening forward: approaching marine biodiversity assessments using acoustic methods

T. Aran Mooney[1], Lucia Di Iorio[2], Marc Lammers[3], Tzu-Hao Lin[4], Sophie L. Nedelec[5], Miles Parsons[6], Craig Radford[7], Ed Urban[8] and Jenni Stanley[1]

[1]Biology Department, Woods Hole Oceanographic Institution, 266 Woods Hole Road, Woods Hole, MA 02543, USA
[2]CHORUS Institute, Phelma Minatec, 3 parvis Louis Néel, 38000 Grenoble, France
[3]Hawaiian Islands Humpback Whale National Marine Sanctuary, 726 South Kihei Road, Kihei, HI 96753, USA
[4]Biodiversity Research Center, Academia Sinica, 128 Academia Road, Section 2, Nankang, Taipei 11529, Taiwan
[5]Biosciences, College of Life and Environmental Sciences, Hatherly Laboratories, University of Exeter, Prince of Wales Road, Exeter EX4 4PS, UK
[6]Australian Institute of Marine Science, Perth, Western Australia 6009, Australia
[7]Institute of Marine Science, Leigh Marine Laboratory, University of Auckland, PO Box 349, Warkworth 0941, New Zealand
[8]Scientific Committee on Oceanic Research, University of Delaware, Newark, DE 19716, USA

TAM, 0000-0002-5098-3354

Ecosystems and the communities they support are changing at alarmingly rapid rates. Tracking species diversity is vital to managing these stressed habitats. Yet, quantifying and monitoring biodiversity is often challenging, especially in ocean habitats. Given that many animals make sounds, these cues travel efficiently under water, and emerging technologies are increasingly cost-effective, passive acoustics (a long-standing ocean observation method) is now a potential means of quantifying and monitoring marine biodiversity. Properly applying acoustics for biodiversity assessments is vital. Our goal here is to provide a timely consideration of emerging methods using passive acoustics to measure marine biodiversity. We provide a summary of the brief history of using passive acoustics to assess marine biodiversity and community structure, a critical assessment of the challenges faced, and outline recommended practices and considerations for acoustic biodiversity measurements. We focused on temperate and tropical seas, where much of the acoustic biodiversity work has been conducted. Overall, we suggest a cautious approach to applying current acoustic indices to assess marine biodiversity.

Key needs are preliminary data and sampling sufficiently to capture the patterns and variability of a habitat. Yet with new analytical tools including source separation and supervised machine learning, there is substantial promise in marine acoustic diversity assessment methods.

# I. Introduction: the need for assessing marine acoustic biodiversity

The ocean covers more than 70% of the Earth's surface, drives much of its climate, provides substantial subsistence and economic resources, and encompasses our planet's largest ecosystems. The marine habitat is host to a multitude of species that have been harvested, observed and studied for millennia. Yet, we are still only beginning to understand the diversity and richness of the species that occupy this vast space. In many habitats and ecosystems, levels of biodiversity are changing, largely at unknown rates [1]. Biodiversity is directly related to ecosystem function and resilience in stressed conditions [2] and, consequently, the need to quantify, document and monitor biodiversity is of critical importance.

While early biodiversity assessments may have been conducted to study evolution and species diversification [3,4], much of the current rationale for studying the biodiversity of marine ecosystems has been driven by the rapid degradation of habitats and the desire for their conservation [5,6]. Indeed, oceans have seen a myriad of stressors in recent years through different anthropogenic factors, such as ocean warming (figure 1), eutrophication, disease, overfishing and ocean acidification [1,8–12]. Biodiversity indices are vital for identifying trends in species richness, designating areas for protection, quantifying the services and value of ecosystems, and highlighting ecosystems that may be vulnerable to a rapidly changing environment [13–16].

Early marine biodiversity assessments often took a broad, global view, usually in an effort to understand the overall trends in the ocean [17,18], and while we are constantly increasing our knowledge of the ocean, we often lack repeated biodiversity estimates for specific habitats. Consequently, it is difficult to accurately characterize changes in the biodiversity of ocean communities. Given the ecological, social and commercial value of marine biodiversity [19,20], it is important that we improve and expand our estimates of marine environments, including adapting to, or leveraging, specific challenges found in the ocean. Ideally, these methods are: (i) correlated to the ecological functions to be measured, (ii) translatable for management applications, (iii) robust to the natural variability of the ecosystem, (iv) reliable, and (v) replicable across multiple spatial and temporal scales [21].

Estimating biodiversity and keeping track of trends in the marine environment is a challenging task, impaired by the salty, aqueous, high-pressure and often light-limited conditions the ocean presents to those studying it. Conventional survey methods, such as visual and photographic/video surveys, are largely adapted from traditional terrestrial methods, yet they allow for only intermittent assessments of biodiversity and are only achievable when sites can be accessed (due to weather, depth and remoteness) and conditions are amenable to visual observation (e.g. not restricted by turbidity). Instruments that provide real-time data on the physical environment (e.g. temperature, light, salinity, etc.) do not directly monitor the diverse biological components of marine ecosystems. Thus, there is a need for marine biodiversity tools to circumvent these challenges to provide rapid assessments of habitats and communities.

Monitoring the sounds present in marine habitats could be a novel and effective way of measuring biodiversity [22,23], providing a complementary tool to other assessment methods. This is because the acoustic environment can be characteristic of a location, reflecting the behaviour of its contributors and inhabitants [24]. The resulting 'soundscape' [25,26] comprises three components [27]: biological sounds (biophony), geophysical sounds (geophony; natural abiotic sources such as wind, waves, rain, water flow, ice cracking) and anthropogenic sounds (anthropophony; human-generated, non-biological sound). 'Soundscape Ecology', more recently termed 'Ecoacoustics' [28], has shown great potential in describing environmental characteristics and detecting change within them.

The diversity of biological sounds seemingly lends itself to acoustic diversity metrics. Species' acoustic signals typically have distinct characteristics, such as frequency (pitch), amplitude, duration and repetition rate. This specificity of sounds can allow for discrimination of species, sex and behaviours. Variability in communication signals have been driven by natural selection as population, species and individuals seek to differentiate themselves or separate themselves from ambient and biological 'noise'. One of the fundamental assumptions about acoustic indices, making them appropriate for monitoring biological sounds, is that most biological sounds have distinct frequency and temporal properties which limit communication overlap [29]. In terrestrial environments, acoustic conditions, such as a full and diverse sound spectrum, can be a predictor of biodiversity and species

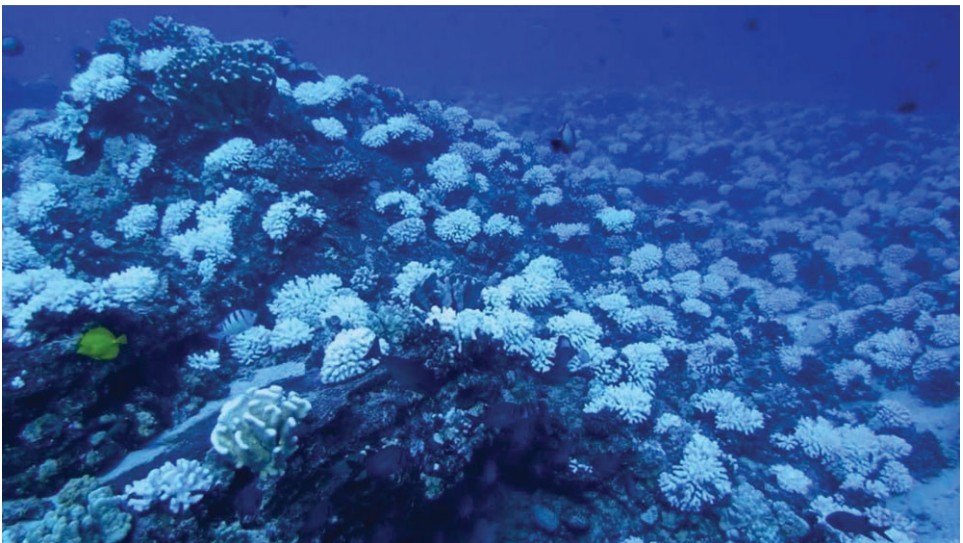

**Figure 1.** Molokini reef, Maui, Hawaii, USA, in October 2016 during a prolonged warming event with extended coral bleaching evident, potentially stressing the local community (Photo: M. Kaplan) [7].

richness [22,30]. For example, models of simulated terrestrial sounds that were tested in an arboreal site suggested that the forest ambient soundscape varies with biological diversity and abundance and that soundscape complexity increases with species richness [31]. Further, the acoustic presence and abundance of certain grasshopper species in a grassland biotope have been shown to be good predictors of biotope quality [32]. One investigation of the community soundscape in a Costa Rican forest revealed that acoustic diversity strongly correlates with vertical forest structure complexity, which suggests that acoustic monitoring could be an effective method of identifying forest patches containing high species diversity [33]. Similar studies have been proposed for freshwater systems [34]. Examining the sounds occurring in remote marine habitats could serve as an effective proxy for estimating habitat quality and community biodiversity, tracking biological processes and detecting natural and changing patterns of biological activity across multiple spatial and temporal scales.

Many metrics have been proposed for measuring species richness in terrestrial environments through acoustic analysis, particularly for forest birds and insects [22,33,35–39]. The role of these soundscape 'indices' is essential to quantify the complexity of the soundscape in a single (or handful of) value(s) [28]. Their success has been variable and their repeatability should be assessed, but it seems that a combination of acoustic indices (rather than a single metric) is more effective at predicting bioacoustic activity [39]. Sueur *et al*. [16] provided a description of the two predominant types of within-group (α) and between-group (β) indices (where the sample unit to be compared is considered to be the group. That group may be a site, habitat or an event in time. In addition to indices, initial attempts have been made to autonomously extract and count ecoacoustic events in a formalized way and to investigate spatio-temporal patterns in these events in the terrestrial environment and freshwater habitats [39–41]. While this does not provide an index of the biodiversity, it does provide a better understanding of its contributors, based on these detected events.

## 1.1. Why ocean acoustics?

The marine environment is rich with sounds (figure 2). As with the terrestrial world, marine biotic sounds reflect many vital biological processes, such as spawning [43–45], courtship [46], feeding [47], social cohesion [48,49] and competition [50] among many species of marine mammals, fishes and invertebrates. Further, sound travels efficiently through water [51], especially compared to light, the basis for many visual and traditional survey methods. Sound itself consists of two components, sound pressure, the compression- and rarefraction-induced sound waves and a scalar quantity typically measured in micropascals; there is also acoustic particle motion, the back-and-forth vibratory nature of sound. Particle motion is a vector thus inherently directional and typically measured in acceleration or velocity. Researchers have typically measured sound pressure and that is what we refer to here, unless otherwise stated. Acoustic signals can be detected over ranges of tens to thousands of metres,

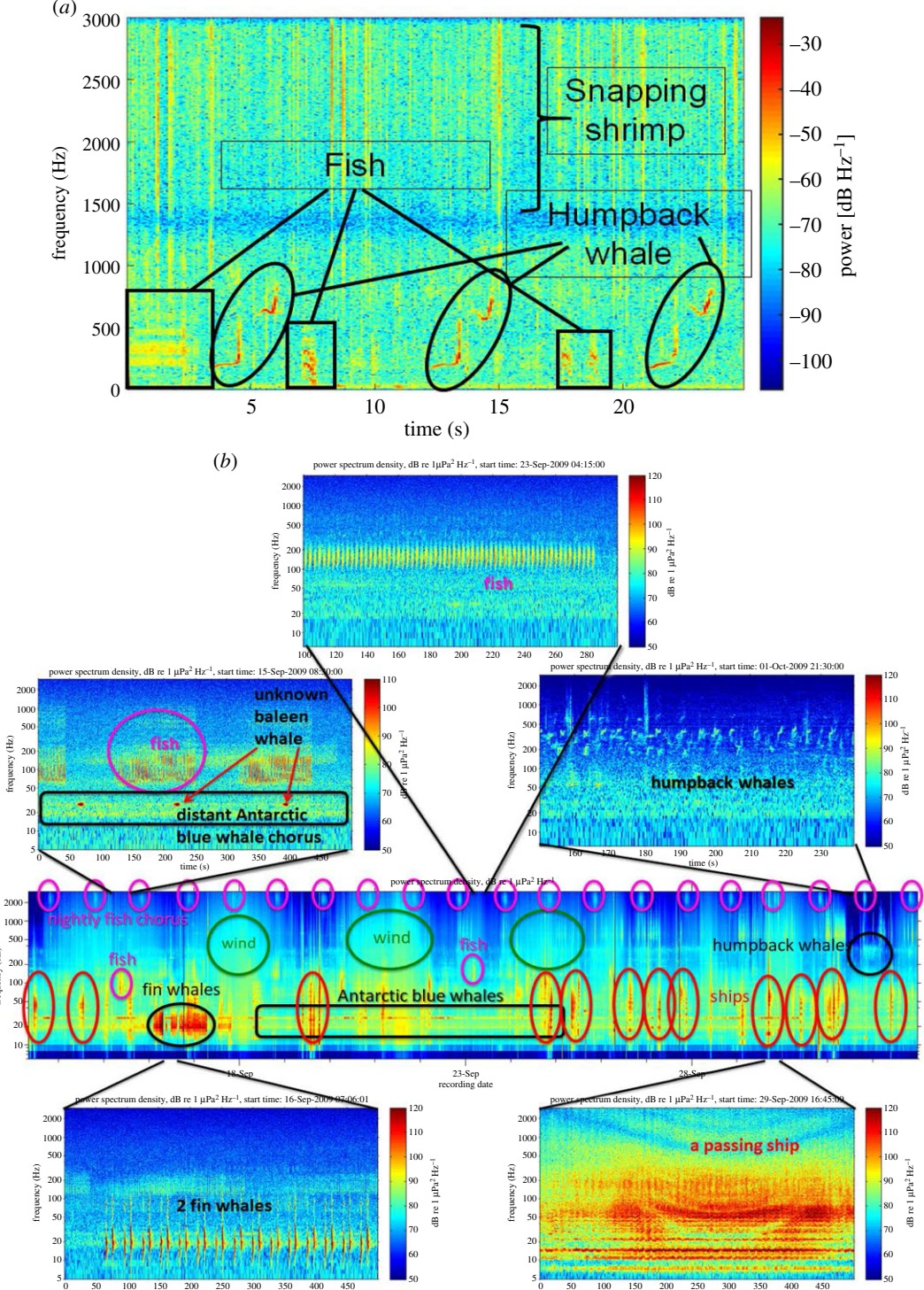

**Figure 2.** (a) Underwater soundscape of Tektite coral reef at the US Virgin Islands showing a diversity of sounds found on that reef (fish, snapping shrimp and marine mammals). (b) Underwater soundscape at the IMOS Perth Canyon site, WA. Middle panel shows a three-week spectrogram. The sounds of fish (individual fish sounds and regular night-time fish chorus), Antarctic blue whales, fin whales, humpback whales, an unidentified baleen whale, wind as well as passing ships are labelled. The five panels around the middle panel display short-term spectrograms of a few example sounds. Modified with permission from [42].

depending on the species producing them, the background ambient noise level and the propagation conditions [52–55]. Soundscape recordings may be collected in a range of conditions when vision and other observing methods are limited, including at depth (e.g. the aphotic zone), during dark hours and in murky waters [56,57]. Further, there is now a suite of advanced passive acoustic recorders that

allow broadband, long-term recordings in a relatively cost-effective manner, greatly expanding their user base and application. Also, they allow locations to be monitored for months or longer without human presence or disturbances. Long-term recordings may be particularly useful to compare biodiversity levels before and after events, such as storms (hurricanes [58,59]), habitat degradation (oil spills, construction [60]), climate-related changes (temperature rise [18]) and in areas of marine protection [61].

Numerous studies have leveraged bioacoustics to improve our understanding of previously unrecognized species presence or abundance. In the ocean, Earth's largest animals can be cryptic. For example, the North Atlantic right whale (NARW; *Eubalaena glacialis*) mothers and calves were generally known to migrate between (typically spring, summer) feeding grounds in the Gulf of Maine and the Bay of Fundy to (winter) calving grounds off the coast of Georgia and Florida. The regions between have long been considered a migratory corridor, with limited NARW presence. Yet, two recent passive acoustics studies have revealed NARW detections in Virginia and the mid-Atlantic in every month of the year [62,63]. These studies demonstrated increased seasonal occurrences in autumn and late winter/early spring, and not just during limited periods of the year. Thus, acoustics is leading to the redefining of habitat use and management of a critically endangered species, particularly as these areas see increased naval activity and use for offshore windfarms and resource extraction. In another example, the Northwest Hawaiian Islands were recently noted as having persistent winter occurrence of humpback whale (*Megaptera novaeangliae*) song. This suggests this area is a previously unrecognized or newly recolonized breeding area for these megafauna [64].

Passive acoustics has helped reveal the presence of essential fish habitat. Sciaenids of the southeastern United States are a multimillion dollar fishery. Luczkovich *et al.* [65] have used moored recorders to identify the spawning areas and timing of multiple Sciaenid species around North Carolina estuaries. Glider-based studies have been used to find new spawning areas of red grouper (*Epinephelus morio*), toadfish (*Opsanus* spp.) and other species [66]. Knowledge of spawning areas allows for the regulation of fishing effort, protection of essential spawning habitat and monitoring of spawning stock biomass fluctuations.

For cryptic species, passive ocean acoustics has a vast advantage for species detection. For example, the addition of acoustics to a survey of a Mediterranean marine protected area (MPA) yielded the detection of a dense population of cusk-eel, *Ophidio rochei*, and suggests this is a reproduction area [67]. This acoustic detection is even more impressive considering that visual census surveys of the fish fauna were carried out for decades on a monthly basis in this MPA, yet they consistently failed to detect the presence of this species. While overlooking cryptic taxa in visual assessments may not be surprising, it is at least striking in the case of snapping shrimp. Composed of two crustacean families (Alpheidae and Palaemonidae), comprising several hundred species, snapping shrimp make their sound through the collapse of a cavitation bubble generated by the closing of their large claw [68,69]. This bubble implosion creates an extremely broadband (many-frequency) sound with sound levels of *ca* 190 dB re 1 µPa. [70]. Snapping shrimp choruses can easily be heard by divers and through boat hulls, but living cryptically in goby holes, oyster reefs or deep in the spongocoel of select sponges, these shrimp are rarely seen. Acoustics is certainly their best means of detection.

Given this apparent bioacoustic richness and the advantages of marine bioacoustics measurement, it has been suggested that the terrestrial community-based methods of relating acoustic and standard biodiversity measures could also be applicable to marine environments [71–75].

To date, much of the work seeking to leverage acoustics to measure marine biodiversity has taken place in areas of concern or biodiversity 'hotspots', areas of high biodiversity or with high rates of endemism. These have often been temperate coastal zones and tropical coral reef habitats. Yet, acoustic diversity has been noted in other key areas, such as polar regions [76–80], albeit to a more limited extent. Marine biodiversity is vital in other areas, such as hydrothermal vents, where acoustic monitoring could be useful. These habitats are certainly regions of interest when monitoring changes in biodiversity and habitat; however, for this review, we focused on temperate and tropical regions, where most of the initial bioacoustic diversity work has been conducted.

Coral reefs are often considered an example of high-biodiversity ocean areas and are frequently referred to as the 'rainforests of the sea' because their productivity and biodiversity rival those of their terrestrial counterpart. Coral-based ecosystems host some of the highest diversity of life per unit area on Earth, and harbour about one-quarter to one-third of all marine species [81–83]. Reef-associated animals are a major source of protein for millions of people. Also, reef habitats offer shoreline protection for communities and are a significant source of tourism revenue. In all, coral reefs are a multi-billion dollar resource, especially for developing countries that have no other major industries [84]. Despite these vast ecosystem services, these biodiversity hotspots are under threat from rapid anthropogenic change and thus coral reefs are also a priority for timely and cost-effective means of

assessing biodiversity and identifying key areas of protection [85]. Ecoacoustic analysis tools may allow us to detect change and monitor the efficacy of management.

Beyond biodiversity, there is a growing understanding that anthropogenic noise may be a stressor in marine environments. Noise impacts are diverse and can influence animal behaviour, physiology and community interactions, and there is growing recognition of the need to address, understand and mitigate many of these impacts. The *International Quiet Ocean Experiment* (IQOE) grew out of this recognition, forming an international programme of research, observation and modelling to better characterize ocean sound fields and to promote understanding of the effects of sound on marine life [86]. The authors of this paper are the IQOE's 'Working Group on Acoustic Measurement of Ocean Biodiversity Hotspots' (see electronic supplementary material for terms of reference). The IQOE was developed because, in most instances, our current knowledge of the specific effects of anthropogenic sound on marine life is inadequate. This scientific uncertainty has made it difficult to balance the need for precaution in protecting marine ecosystems against the potentially large costs to socially important noise-producing activities, such as commercial shipping, offshore energy exploration and development, and military readiness. A key component of understanding noise impacts is to study the natural ocean soundscape, how animals use this soundscape and how soundscapes are changing. We may then be able to leverage patterns in those soundscapes to better understand the ocean.

The goal of this paper is to provide a timely consideration of the emerging methods that use passive acoustics to measure biodiversity and recommend avenues forward in this developing field. We provide a summary of the brief history of using passive acoustic methods to assess marine biodiversity and community structure, and a critical assessment of some of the challenges faced. A key aim of this paper is to outline recommended practices and considerations for assessing marine acoustic diversity and by extension, biodiversity. We hope such efforts will allow researchers, and potentially managers, to apply acoustics for conservation biology needs, such as comparing biodiversity at different locations within a habitat type or region (i.e. which reef to protect or study) and to quantifying temporal trends at a given location (i.e. identify biodiversity loss over time). Given that this is still an emerging field, we describe future research needs to ultimately improve and broaden biodiversity estimates and tracking using acoustics.

# 2. Initial methods and challenges in assessing aquatic acoustic biodiversity

## 2.1. Early work

Research groups studying terrestrial ecosystems using ecoacoustics have led the way in applying acoustic indices to measures of biodiversity. One of the early ideas promoted by these groups was to use recordings and acoustic indices to make biodiversity assessments in a rapid timeframe [30]. These included the acoustic entropy index ($H$), a calculation adapted from the Shannon's diversity index, which aims to reflect the evenness of a signal's amplitude over time and across the full range of frequencies [30]. Acoustic complexity index (ACI), originally developed for analysing avian communities, summarizes fluctuations of sound within frequency bins across time, then summarizing these variations for the whole frequency range [87]. These and other indices have been reviewed by Harris *et al.* [88] and Buxton *et al.* [39].

Coral reefs have provided a range of opportunities and studies to begin linking and testing the efficacy of bioacoustic diversity indices to more traditional observations. Early work suggested sound pressure levels were associated with coral cover, showing promise in using acoustics to assess the community [89], but the recordings were short duration and limited to simple acoustic measures (broadband and octave-band sound pressure levels). Additional work in temperate environments showed promise, addressing three patches of very different habitats (mud, gravel and rocky cliff) recorded for short periods of time [71]. Differences were found between the sites, using terrestrial-derived metrics of acoustic complexity and acoustic diversity, but the authors attributed these results to different levels of snapping shrimp; they did not assess biodiversity. Parks *et al.* [74] found another use for one of the indices, acoustic entropy, to identify calls of mysticete whales, after the background noise was filtered and reduced. The authors indicated that the methods had promise but did not actually link acoustic records to biodiversity assessments. Similarly, polar researchers have used acoustic metrics, often to quantify marine mammal-based acoustic diversity [77], listening across very large distances. From these key stepping stones, more comprehensive measurements were needed.

## 2.2. Challenges posed by variability

One important consideration in assessing biodiversity is the sampling regime [90,91]. A soundscape sampling regime's effectiveness is influenced by natural acoustic variability. Substantial work has shown that the sound levels and frequency content of coral and temperate reef soundscapes vary substantially in space and time [56,92–95]. Changes in the sound field, including fish and snapping shrimp sound production, can change with a variety of factors, including season, lunar periodicity, lunar light levels, temperature, upwelling, tides, salinity and time of day [96–100]. The activity of the animals present affects their detectability; in acoustic assessments, animals can only be detected if they are acoustically active. Conversely, some animal signals will swamp those of others. For example, marine mammals are often dominant contributors to soundscapes from polar to tropical habitats. Calling patterns vary from daily to seasonally, and have substantial spatial variability [101,102]. In some low-latitude sites, fish and invertebrate sounds that are easily detectable in the summer are overwhelmed and masked by the persistent song of humpback whales in the winter [7].

Even within a habitat or community, soundscapes can vary. Kaplan *et al.* [94] showed that within a reef small, yet significant, variations exist at recording stations just metres apart. Within a reef area, there are areas of preference, and they may become acoustic 'hotspots' as seen in concentrations of snapping shrimp in Hawaii [103]. Fish choruses and calls can also vary substantially across adjacent habitats [104] and by geographical location [105]. One reason for these within-reef differences is probably driven by the sheer statical power of near-continuous acoustic records [94]. It is possible to find differences but they reflect the within-habitat variability. The range (or general extent) of this variability seems less than the differences among dissimilar habitats, but we are only beginning to understand the extent or patterns of such 'intra-habitat' heterogeneity [106]. For example, a key area of study would be to address discerning reef health and diversity gradients from acoustic indices. It is important that both within- and among-habitat differences be addressed to properly use acoustic diversity metrics.

A research frontier for the field of ecoacoustics is, therefore, establishing data collection requirements and sampling regimes that are appropriate for the variability they attempt to quantify. Different goals may be served by different sampling regimes, just as visual surveys of transient and cryptobenthic fish have different standard methodologies.

## 2.3. Comparing soundscapes to other biodiversity measures

Monitoring underwater soundscapes to assess community structure requires comparative observations. Staatermann *et al.* [72] compared acoustic data to visual surveys in their study area and found a relationship between the 'low band' frequencies (less than 1000 Hz) and the richness and abundance of cryptic fish. This lower-frequency band is often indicative of fish, but not invertebrate, sounds. Kaplan *et al.* [94] collected four months of acoustic data in the Caribbean Sea and compared them with visual surveys of fishes and benthic cover. They found a relationship between crepuscular increases in fish chorusing to fish abundances and coral cover. Notably, snapping shrimp patterns (1.5–24 kHz) were not associated with either metric. A 16-month follow-up study at additional Pacific Ocean sites showed similar patterns [7]. Desiderà *et al.* [107] found a relationship between sound types (not indices) and taxonomic diversity. They compared years of visual census data from a Mediterranean Sea MPA to acoustic data and found a strong relationship between taxonomic diversity and acoustic diversity (but not acoustic indices).

In fact, many marine studies have shown that there is not a clear relationship between biodiversity and the terrestrial-derived acoustic indices. With respect to $H$ and ACI—two prominent metrics—snapping shrimp sound patterns and other dominant biological sounds seemed to greatly affect the power of the index, suggesting that loud, frequent or omnipresent sounds could bias some indices. Kaplan *et al.* [94] calculated acoustic entropy ($H$) and acoustic complexity (ACI) from their recordings and found no relationship to fish abundances (discerned using lower, fish-dominated frequencies, snapping shrimp-dominated high frequencies and across the full recording bandwidth). Buxton *et al.* also found that acoustic indices did not reliably predict bioacoustic activity in marine habitats. Suggesting such a result was due to the overlap of many biological signals with both the snapping shrimp and anthropogenic sounds [39]. Staatermann *et al.* [72] found a similar effect with index values being dominated by the presence and intensity of Bocon toadfish (*Amphichthys cryptocentrus*) or snapping shrimp. This supports early work by McWilliam & Hawkins [71], testing these indices in environments that were dominated by snapping shrimp, finding differences between sites, but suggesting that the differences were due to snapping shrimp patterns, not overall community presence.

Indeed, snapping shrimp are present and abundant in healthy and 'degraded' reefs, but their relative extents in both habitats are not yet well understood [56,108]. As some species of snapping shrimp are often found in sponges or symbiotically living with gobies, it is possible that their snap rates and levels may be associated with those of animals and communities [109,110]. However, acoustic diversity metrics heavily based on the sound production by these persistent and acoustically dominant crustaceans or other mass phenomena, such as fish choruses, may ultimately be limited in their ability to measure biodiversity [111].

These studies contrast somewhat with work from Moorea that appeared to find a relationship between terrestrial acoustic indices and visually observed biota [112,113]. However, a limitation in those efforts may have been that they were relatively short-term recordings that were also made sequentially, not in parallel, which could overlook or be influenced by the acoustic variability within a site, or temporal variability of a diel or lunar cycle. Elise *et al*. [21] made similar claims regarding the effectiveness of six acoustic indices. However, they also sampled for a very short duration (hours to 24 h) remarkably concluding that listening for only 2 h could be used to discriminate sites.

Making conclusions about community complexity from snapshot recording is potentially misleading, given the substantial variability of coral reef soundscapes. Notably, the efficacy of some indices is greatly affected by temporal variability of habitats [88]. Further, recent work by two research groups, working at sites in the Mediterranean Sea, Bahamas and Pamlico Sound, USA, evaluated the usefulness of several acoustic indices. Using *in situ* observations, and in the case of [114], also testing $H$ and ACI against synthetic sounds, the indices were strongly affected by temporal variations in the activity of single sound-producing species (such as snapping shrimp or a loud fish chorus). The ACI was found to be sensitive to variations in both sound abundance and sound diversity, making it difficult to discern between these variables. Call type, calculation parameters and the resolution of the analysis also affected the indices' performance. In these cases, the authors concluded that 'ACI and $H$, therefore, cannot be assumed to track call diversity, and the utility of these metrics as biodiversity indicators in marine environments may be limited' [111,114,115].

# 3. Considerations for using acoustic methods to estimate marine biodiversity

While the ultimate goal of soundscape metrics is to provide a measure of the local ecosystem's state and/or biodiversity using the characteristics of the biological sounds, it is important to note that these may not be directly related. While soundscape metrics may be valuable, the variety and variability of life at a recording location may not be directly reflected by the variety and variability of acoustic signals (figure 3). For example, not all sounds are biological in origin, not all animals will make sounds and there is significant variability in sounds that are produced. Soniferous species produce different varieties of sounds and can do so at different rates. Not all species produce sounds at the same intensity and, depending on habitat, not all sounds travel to the recording device in the same manner. Defining each of these factors is difficult. In the same way that key indicator species can help shape understanding of ecosystem health [116], there may be a small number of species that dominate the soundscape, mask other sounds and impact acoustic diversity assessments. Yet, their presence could be an indication of high biodiversity of non-vocal species. Understanding these dynamics and the biases they create is vital to soundscape measurements. It is imperative that we evaluate the biases of methods of biodiversity estimation and ensure that the limitations of methods are taken into account when they are used.

These soundscape challenges are not unlike those of other underwater biodiversity assessment methods (e.g. trawls, eDNA, diver, etc.). For example, visual-based assessments (such as baited remote underwater video) can be impaired by the variability of the observed cues and the sampling methods. Several methods have been developed to determine how the behaviour of the animal contributes to it being observed. This includes addressing the effects of diver presence on fish behaviour [117] or bait plumes on the sample area [118], and how the maximum number of animals observed represents the relative or absolute number of animals in the area, such as 'MaxNo' or 'MaxN' used in baited remote underwater video surveys [117,119,120]. Many key ecological parameters vary substantially [121], thus short-term and low-frequency sampling can miss this variability and potentially lead to mischaracterizing a habitat. The uncertainty is partly addressed by longer observations, data from multiple sources and ground-truthing of methods. In developing representative acoustic metrics, there are several considerations that affect how the data must be approached to tease apart how acoustic diversity and biodiversity are related.

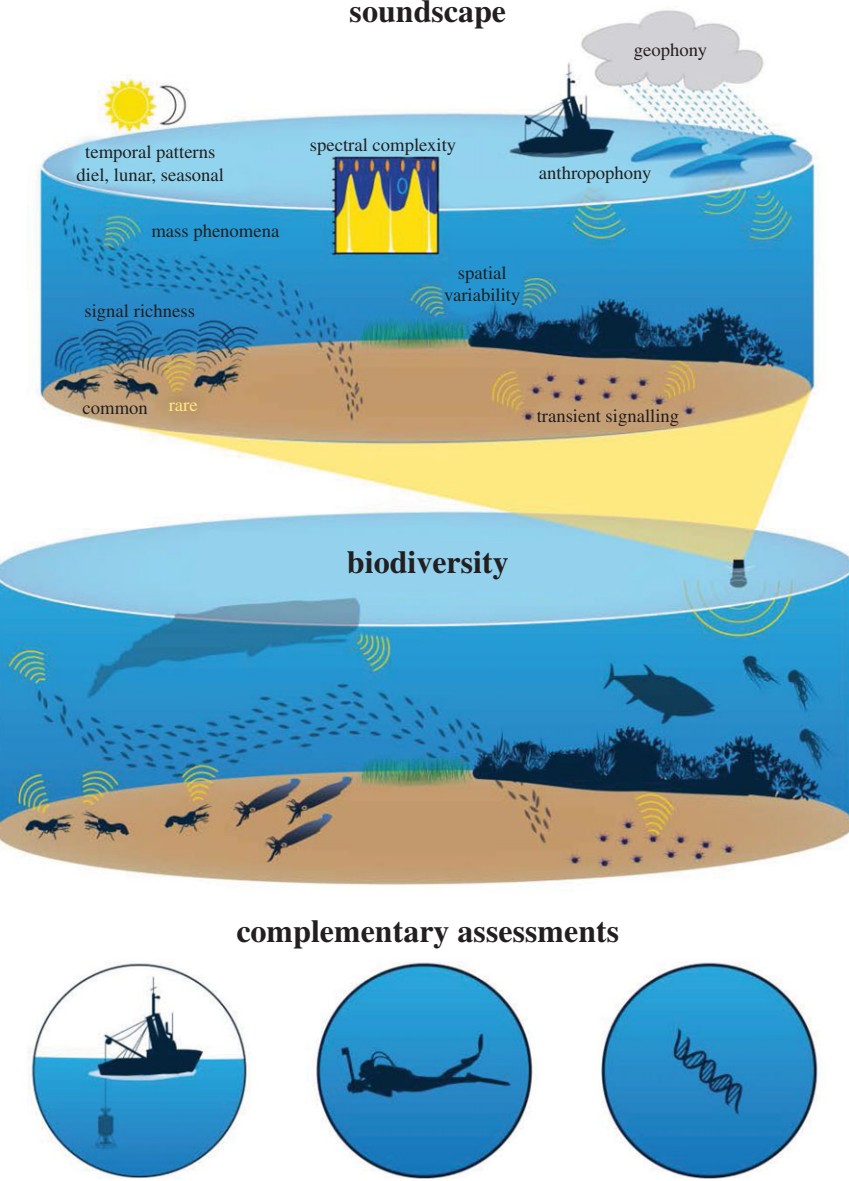

**Figure 3.** Assessing biodiversity from the ocean soundscape. Diversity of bioacoustic signals in the soundscape is one way to estimate biodiversity. Complications to bioacoustic diversity measurements include the variability of the soundscape including its bioacoustic cues, presence of geophysical and anthropogenic 'noise', propagation and transmission loss in shallow water habitats, and a bias towards species that produce sounds.

## 3.1. The end user

The needs of the end user are the ultimate driver for the design of any acoustic metric [122]. There are potentially multiple users of acoustic diversity information. Managers and scientists may hold vastly different expectations of what information an 'acoustic index' will provide. This may range from the marine park manager looking to compare biological complexity across multiple sites, to the scientist attempting to identify the point at which an entire coral reef suffered a bleaching event, to the fisheries manager trying to confirm location and timing of spawning fish. The sound types, local conditions that can feed into the index and the spatio-temporal scales over which the index will be relevant, all affect the potential design and outcome of a monitoring programme. Yet, for management of declining resources, properly evaluating a community's health and biodiversity is critical, underscoring the need for proper evaluations. The users' needs (and limitations) will affect the sampling regime, and they must balance sufficient evaluation of the community with rapid reporting for conservation needs.

## 3.2. Separating biotic, geophysical and anthropogenic sounds

Any assessment of the soundscape requires segregation of the three predominant source types: biological, anthropogenic and natural abiotic sounds. Separation of these three contributors should be an initial step of any soundscape evaluation. Some analyses may aim to assess and compare the overall soundscape contributions of biological and anthropogenic sounds [123,124], or to assess the influence of abiotic sounds (e.g. wind and wave noise) on biological functions [125]. For almost any soundscape analysis, it is necessary to segregate the contributors into these functional groups, and 'noise' sources can clearly affect acoustic metrics [39]. It is essential to isolate anthropogenic and geophysical contributions to the soundscape, before acoustic measurements of the biological sources can be linked with local biodiversity.

Accurately segregating anthropogenic and geophysical sounds in long-term passive acoustic data is non-trivial and has been attempted many times, with varying degrees of success (e.g. [126–129]). It has been suggested that biological sources, which are generally transient signals, exhibit higher acoustic variability than anthropogenic sources [127]. However, several anthropogenic sources, such as the passing of large vessels, bear similar acoustic variability to biological sources, particularly mass phenomena, such as fish choruses, and can be mistaken as such. The biotic and abiotic sources may be separated by acquiring recordings that include the commencement and cessation of the source [126]. There may also be a need to remove entire periods of data where anthropogenic noise (e.g. seismic air-gun firing, dredging, shipping channels or pile driving) or geophysical noise (e.g. rain, current flow, wind-driven waves) partially or completely masks even the loudest of biological signals ([104,130], figure 2). Other geophysical sounds such as ice-generated noise can be seasonally persistent, high amplitude and similar to biological sounds [131,132], making it easy to detect but difficult to distinguish from biological sounds.

## 3.3. Key characteristics of bioacoustic diversity

Relating a location's bioacoustic diversity to biological biodiversity is challenging for a number of reasons. Animals produce sounds intentionally (e.g. communication, orientation, foraging) or non-intentionally (when feeding, moving, etc.) [133–135], with varying information content in different sounds. While many sources of these biological sounds have been identified, a larger proportion are still unconfirmed. All, however, contribute to the diversity of the soundscape and provide important information on the location's biodiversity [107]. These calls may be produced rarely, with no temporal pattern, or in series, often with a distinct temporal pattern, depending on their function. Marine mammal signals are immensely variable, with cetaceans and pinnipeds being the most soniferous. With respect to frequency content, mysticetes are typically confined to lower frequencies (less than a few thousand hertz), odontocetes spanning a few kHz to 100–200 kHz and pinnipeds (including the acoustically diverse Weddell seal (*Leptonychotes weddellii*)) ranging from 80 Hz to 24 kHz [136,137]. Most sounds produced by fishes are limited to a small frequency range, tens to a few thousand hertz (e.g. *ca* 5000 Hz and lower) [138,139]. There are some marine invertebrates, such as the American lobster, *Homarus americanus*, that also produce low-frequency sounds [140]. Yet, many invertebrates occupy a much broader acoustic range; many start near 2000 Hz and extend above 50 kHz [55,141,142], with snapping shrimp sound energy extending out to 200 kHz and beyond. Marine biodiversity analysis must distinguish between these sounds that might represent a species, and potentially also sounds that are more general or difficult to identify to species (i.e. snapping shrimp and some delphinids).

### 3.3.1. Mass phenomena

Frequently emitted sounds can form mass phenomena referred to as choruses [102,143]. Mass phenomena are the grouping of multiple sounds that are no longer discrete signals; rather they have coalesced into a single broader signal, of significantly longer duration and potentially different spectral content. This may occur when sounds from singing whales, a large school of soniferous fish or invertebrates converge, forming a continuous chorus 'when the noise from many individuals is continuously above background for an extended period using an equipment averaging time of 1 second', with a significant increase above background (greater than 3 dB re 1 µPa, [143]).

The choruses and song of some cetaceans and pinnipeds have been studied for decades [79,101,102,144]. Their signals are high amplitude and can be detected and localized out to many tens of kilometres [54,145], and may occur for months at a time. Fish and invertebrate choruses may also be intense and dominant in a soundscape [146–149] and across a period of up to several hours or

across breeding seasons [150–153]. A chorus can mask the individual, often lower-level, transient sounds and thus will often reduce the detectability of many signals and the overall acoustic diversity. Regardless, it is pivotal to separate mass phenomena from more rarely occurring sounds when assessing bioacoustic diversity [107], and for these sounds to be considered separately.

### 3.3.2. Transient sounds

Transient sounds are referred to here as all uncommon biological sounds that occur in low abundances or generally do not produce mass phenomena but, nonetheless, are important components in acoustic diversity assessments. Indeed, rare observations, acoustic or otherwise, greatly limit probability of detection and thus raw counts of individuals will be biased towards more gregarious, larger, louder or easily detected species leading towards erroneous richness measurements [154,155]. Further, detecting a high number of rare species is typical of many surveys. Accurately capturing this number is crucial as rarity of species and overall site richness have implications for threat status and extinction risk, and the number of rare species is also used to establish spatial conservation priorities [156,157].

With respect to bioacoustics, many of these sounds may be unintentional cues, such as the transient broadband sounds emitted by some invertebrates (e.g. rasping sound produced by the gastric mill of crabs during feeding). However, invertebrates will also acoustically signal, such as stridulation, e.g. rubbing together of hard body parts [55,158–160]. Such transient sounds are diverse, but as many of them are not used for communication, their specificity is low, e.g. species, individual, etc. [142]. Fish vocalizations for communication are often species-specific and more diverse, but confined to a narrower frequency band compared to transient invertebrate sounds [161–164]. Vocal repertoire can vary significantly between fish species. For example, haddock (*Melanogrammus aeglefinnus*) display a number of calls associated with various spawning behaviours, while Atlantic cod (*Gadus morhua*) are thought to be less versatile vocalists during courtship [43,124,165]. Yet, they may also make unintentional sounds, such as from fast swimming moments in jacks (carangids) or the scrapping of algae off reefs by parrot fishes (scarids) [166,167].

In contrast with fish and invertebrates, marine mammals are known to produce a large variety of sounds, often with species-specific vocal repertoires [168–171]. Mysticetes, beaked whales and pinnipeds display species-specific spectral peaks, which can be identified in spectral statistics and used to facilitate the analysis of the spatio-temporal changes of marine mammal sounds [42,172,173]. Most delphinids produce a diverse tonal sound repertoire that spans a wide frequency range and changes with behavioural contexts [174–176]. Quantification of the frequency diversity may be associated with diversity in both species and behavioural states [177]. In addition, the high-frequency whistle and click sounds produced by toothed whales are highly directional [178–180] and recorded spectra may vary with azimuth, when vocalizing animals change their heading directions. Additionally, noise, both anthropogenic and physical, may be transient and have the potential to influence biological sound production [181]. Thus, the ecological interpretation of the spectral complexity in the analysis of marine soundscapes must be carefully developed.

The diversity of transient sounds will be critical for the evaluation of biodiversity in species and behavioural levels. Manually inspecting the presence/absence of different sound types is time consuming, but necessary for building baseline validation information. The detection of transient biophonic sounds, particularly the rare ones, can be impaired by anthropogenic noises or biological choruses, the assessment of their diversity can be very challenging. Using the directional component in sound inherent in particle motion vectors or the use of a hydrophone array to determine source direction are potential methods of separating transient signals from one another and also to separate them from noise (e.g. [182,183]). Given the lower probability of detecting rare and transient sounds [154,155,184], repeated surveys, multiple methods and longer-term observations are also key methods to improve species detections and quality of richness estimations

## 3.4. Other factors influencing acoustic diversity measurements

### 3.4.1. Sound levels

Sound pressure levels and received acoustic energy are some of the key descriptors used to produce many acoustic biodiversity assessments and indices [16]. Yet, the sounds produced by individuals, species and groups of animals can vary vastly in amplitude. Similar to the effect of mass phenomena, the impact of high-amplitude over lower-amplitude calls can bias an index towards a more diverse

estimate. In Australia, for example, the source levels of mulloway (*Argyrosomus japonicus*) and West Australian dhufish (*Glaucomsoma hebraicum*) differ by greater than 30 dB re 1 µPa [185,186]. In the USA, Gulf corvina (*Cynoscion othonopterus*) and Atlantic cod (*Gadhus morhua*) differ by an even greater level, approximately 40 dB (100×) [124,148]. Within a species, source levels may also vary [187,188]; advertisement calls may be much greater in sound level than ancillary cues, such as sounds associated with movement or feeding. In some fish, source levels can also vary seasonally with changes in sonic muscles related to reproduction [189,190]. Yet for most species, we typically do not know the source levels of many calls, nor can this be easily determined.

In addition, received levels are impacted by propagation conditions of the area, such as shallow versus deep water, bottom type, seabed rugosity and physical geometric spreading. Sounds tend to attenuate faster in shallow waters as they repeatedly reflect and are scattered off the seabed and surface [191]. This attenuation rate tends to be more pronounced for low frequencies and in areas with rougher boundaries (more rugose reefs), although high frequencies will absorb faster. Thus, open water can act as a low-pass filter, enabling better lower-frequency sound propagation. Sound pressure also decreases logarithmically with distance as sound spreads from a source [136]. Consequently, sound levels decrease faster closer to the source (i.e. 20 dB or 10× at 10 m and 40 dB at 100 m). With respect to bioacoustics, the physical conditions, therefore, disproportionally affect the detection of lower-amplitude signals and low-frequency animals that tend to communicate in shallow waters and short ranges (e.g. many fish sounds). Consequently, care must be taken regarding how received levels are interpreted in acoustic biodiversity indices.

### 3.4.2. Spatial range and scale

As noted above, variability in acoustic propagation (predominantly due to bathymetry and water depth) and the differences in source level of biological signals mean that the ranges over which sounds may be measured will vary. Estimating detection ranges, density within an area [154,192], and effective width of observation, as well as modelling detection is difficult.

Sounds from cryptic or quiet species, such as *G. hebraicum* or *G. morhua*, or the damselfish, *Dascyllus albisella*, may only be detected above ambient noise (natural and/or anthropogenic) at ranges of a few to a few tens of metres [46,124,185]. If such species are not widely or uniformly distributed, then their contribution to local soundscapes, and therefore potentially the applicability of that soundscape to estimate diversity, is limited to the detection range. By contrast, high-amplitude cetacean and fish signals generated in deeper water may propagate great distances and dominate soundscapes [54,193,194]. Initial soundscape studies showed differences between soundscapes separated by several kilometres [89,93,195], and more recent efforts have noted that reefs separated by just a few hundred metres may vary [149]. Further, there are bioacoustic differences within a reef (distances of a few metres), suggesting the soundscapes reflect biological hotspots or areas of particular activity [94,98,112]. As a result, spatial replication is a necessity, not only to account for variability between sites, but also to ensure that the soundscapes recorded are representative of an area or habitat.

### 3.4.3. Timescales

Any sampling of acoustic diversity should be conducted over a timeframe that is considered representative of the site and sufficient to capture rare species, which will reduce bias in estimates of species richness and ensure relatively high mean detection probability overall. Selection of an appropriate timeframe depends on the objective of the acoustic metric. Prior to establishing a representative timeframe, it is, therefore, highly preferably to record multiple cycles of the temporal variability that is to be quantified and consider this alongside the goal of the indices to be used. Variations in soundscapes and sound production by individual species have been shown to occur in relation to tidal, solar, lunar (and semi-lunar), seasonal and annual cycles [92,99,100]. Diel patterns and crepuscular acoustic activity are some of the largest sources of variability on a coral reef [7,94]. Local geophysical or oceanographic conditions (e.g. tide-dependent water depth) can change on similar scales, altering the soundscape during their cycle [196]. In addition, habitats may have varying temporal scales of acoustic diversity. For example, habitat A may exhibit the same degree of acoustic diversity as habitat B, but over a very different timescale (e.g. one week versus six months).

It is also not necessarily appropriate to assume that a species produces sound over the same timeframes at different locations. Fish choruses, potentially produced by the same species, have been shown to change their timing at different locations, occurring at different times of the day/night, or displaying different lunar

patterns in intensity [104]. Similarly, snapping shrimp have been shown to produce sounds at different times and daily patterns in adjacent reefs, just hundreds of metres apart [98].

Migratory species, whether marine mammal, fish or invertebrate, can contribute significantly to the local soundscape for the duration of their passing and this can be species-, site- and condition-specific. At the hundreds of metres to kilometres scale, Parsons *et al.* [96,197] proposed that *A. japonicus* are mobile, individually moving slowly along a river calling as they go, and that the aggregation (and therefore chorus) position changes during the course of the evening chorus. Whether this was to match changing environmental conditions (location of high tide edge) was not known. On a shorter timeframe, the sound source, such as a whale, may be mobile while vocalizing [198]. Humpback whales (*M. novaeangliae*), for example, can spend several months around Exmouth Gulf, in Western Australia [199], altering the local soundscape of several areas of the Ningaloo Reef with their song, but pass through Port Hedland (500 km along the West Australian coast) over the course of a few weeks. Migratory species can influence visual surveys in a similar way; therefore, this is not a new problem, but we must consider how we wish to standardize ecoacoustic survey methods in the light of this issue.

The result of these variations is that at a single site, soundscapes (and snapshots of acoustic diversity) can change substantially. Sampling at one time (e.g. sunset or high tide) can be significantly different from later that day, and the same time the next day. With the formation of environmentally driven mass phenomena, the soundscape can change on an hourly basis or quicker. Without confirming which sounds are important for the particular acoustic metric, and targeting times at which these occur, it is necessary to sample at high enough resolution to capture short-term changes and for sufficient duration to encompass as much potential variation as possible [104].

One possible mechanism for evaluating sampling duration is cumulative dynamic range [200], which calculates the variability of sound levels (in power spectral density) as the number of samples increases. As an increasing number of louder and quieter levels are recorded, the mean dynamic range increases in a logarithmic fashion until it eventually asymptotes with time (number of samples). This reflects that the acoustic variation now generally lies between the 1st and 99th percentiles and hence does not contribute to the dynamic range computed. The challenge is that environments must be substantially different (much louder or quieter) to influence the dynamic range with increasing samples, and short-term extreme events, such as passing cyclones, may be missed.

Another possible mechanism for evaluating sampling duration is to examine acoustic data computed over a variety of time windows. Yet, one must first make sure they have sampled for long enough to uncover the variation and signals present, a key step often ignored. This essentially requires longer-duration recordings. Subsampling will often miss the transient (perhaps diverse) events [201].

Pilot data and a short-term evaluation of the sound field allow an initial evaluation of the dynamics and potential sampling regime needed for the acoustic recordings and evaluation. These can contribute to supplementary information and support a study design. In order to fully quantify acoustic diversity, long-term sampling (multi-year) must be optimized to include all potentially significant changes in the soundscape to assess overall diversity. Overall, while potentially attractive, 'snapshots' (e.g. minutes, hours, few days) are likely to miss many of the transient sounds acoustic behaviours present and are not recommended.

Of course, vulnerable habitats such as coral reefs are rapidly deteriorating due to anthropogenic change. Some cyclones [59], disease and bleaching events [202] can impact diversity. Thus, in some cases, there might not be time to base ecoacoustic evaluations on multi-year recordings before they change. Yet not all major (bleaching) events affect acoustic patterns [7], but understanding the natural patterns is key to unravelling such differences. Thus, it seems that given both their strength and subtleties, recording daily, lunar and season variation is vital to accurate acoustic diversity assessments. In essence, a balance must be struck where the limitations and compromises of methods are taken into account in the context of their intended use, but sufficient information is captured to be able to make a valid and robust assessment.

# 4. Current methods supporting acoustic diversity assessments

The evaluation of marine soundscapes as a tool for assessing biodiversity is a rapidly developing field. As such, no single method has yet emerged that is broadly considered the standard for quantifying biodiversity acoustically. Some acoustic techniques that are being explored and used are outlined below, though most of these signal processing and data visualization methods are continuing to evolve. Emerging quantitative approaches offer a means to address key needs in this area and are addressed below.

## 4.1. Long-term spectrogram analyses

Long-term spectrogram analysis (LTSA) is a common method to investigate the spectral–temporal change of marine soundscapes. By measuring the spectral average in a user-defined time window (e.g. 5 min interval), a long-term spectrogram can visualize the presence of various sound sources. Choruses are particularly visible in long-term spectrograms and power spectra as they represent energetic mass phenomena [203,204].

It is also possible to take different statistical measurements of spectral variation, such as the spectral median or a specific percentile, to emphasize changes in the sources of continuous sounds, such as environmental sounds and biological choruses. Considering only high percentiles (95th or 99th) within specific frequency ranges can be used to characterize transient high-energy sounds [205]. Collating long-term spectrograms into a single power spectral density probability (PSDP) plot [206] can provide an overview of the levels of energy likely to occur at a given frequency and are source-specific if the source is significant at a site. However, this approach can only display the rough spectral pattern of transient sounds and cannot precisely display the time-dependent spectral modulation of transient sounds. Thus, a combination of manual inspection and automatic detectors/classifiers may still be necessary for investigating the structure of the 'rare' sounds. Perhaps one way to use LTSA processing to address these rare sounds leverages the 'constraint' that many marine soundscapes have, a predominance of lower-frequency sounds and thus non-Gaussian and right-skewed distribution of power spectral densities (PSDs). The difference between the mean and median or the background ambient noise PSDs can be effective to reduce the influence of continuous signals and thus enhance the presence of transient signals [131,207] (figure 4).

## 4.2. Clustering of audio events

In the analysis of long-duration recordings, we generally do not have sufficient time and personnel to listen and manually scroll through substantial durations of audio data. A technique that can reveal the structure of long-term recordings is necessary to identify the factors driving changes in acoustic diversity. Clustering is a common approach in information retrieval [208]. It assumes that a dataset exhibits an unknown structure. The dataset can be broken down and assessed by finding groups that have similar and unique characteristics.

To facilitate the evaluation of acoustic diversity, clustering algorithms can be run on a long-term spectrogram. By grouping audio clips with similar spectral characteristics, a long-term spectrogram can be summarized into a limited number of audio clusters to facilitate the evaluation of acoustic phenology, such as diurnally and seasonally changing patterns [207]. On the basis of audio clusters, it is possible to measure acoustic diversity and evaluate the relationship with biodiversity (figure 5). However, the clustering performance may be biased when spectral representations are corrupted by non-biological sources [210]. Another option is to integrate soundscape metrics that are less sensitive to unwanted noise with clustering to improve the content interpretation of long-duration environmental recordings [211]. Despite this, there is still a need to examine the sounds that contribute to each audio cluster in order to understand the ecological factors that influence changes in acoustic diversity (e.g. the likely sources, sound production rates, paired with physical process, etc.)

## 4.3. Acoustic activity detection and feature extraction

Rule-based detectors and feature extractions are widely employed in the assessment of the acoustical behaviour of soniferous marine animals. For example, the sounds of many fish species, including Sciaenidae (croakers and drums), are mainly detected as a sequence of pulses [212,213]. The temporal patterns, or rhythms, which can be compared to acoustic barcodes, contain information for discriminating sound types and eventually species. Thus, analyses of a signal's temporal pattern can be very effective for evaluating the behaviour and species composition of soniferous fish [214], though this can be non-trivial if the species have large vocal repertoires [165,152]. Key advances from marine mammal research, such as spectrogram correlation and parametrizing and grouping spectral peaks, may provide some important tools [173,215].

Another example of rule-based detection and feature extraction is the contour tracking of marine mammals' tonal sounds [216–218]. The structure of tonal repertoires can be investigated by grouping different clusters with similar modulation of representative frequencies on a contour. The determination of the number of clusters may be a challenge. Instead of evaluating the cluster number manually, it is

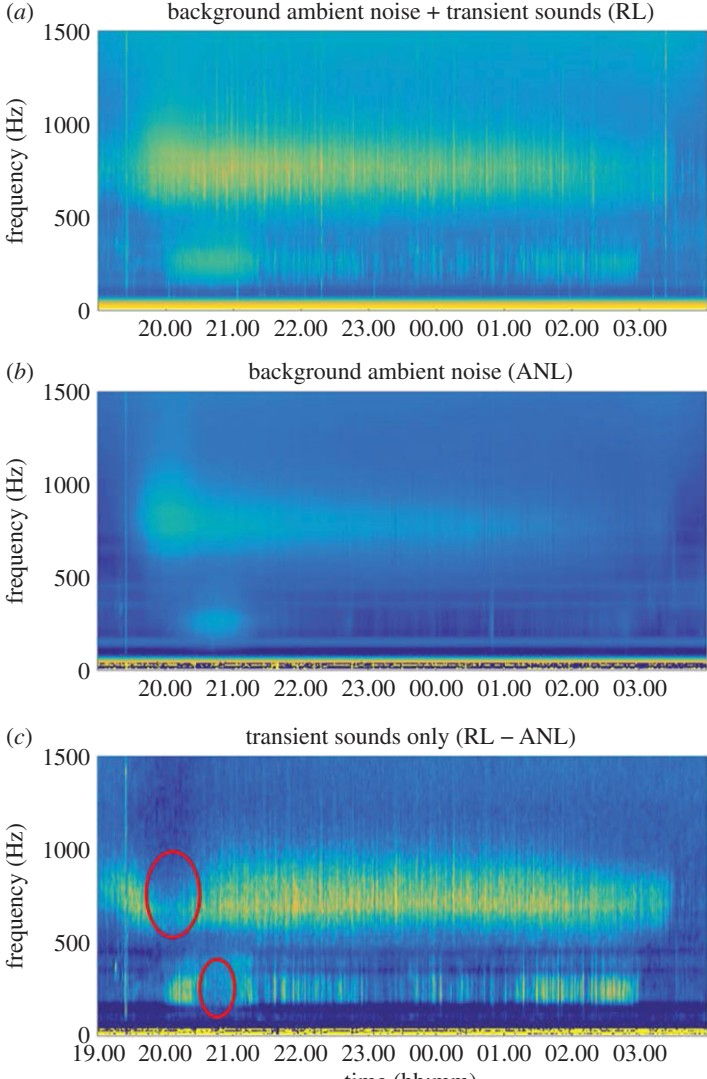

**Figure 4.** Eight-hour spectrograms of the lower-frequency component of a Mediterranean soundscape. (*a*) The classical overall spectrogram with combined background ambient noise and transient sounds, while (*b*) shows the spectrogram of the background ambient noise only, without individual transient sounds [131]. Bright green areas indicate fish choruses. (*c*) The spectrogram of the difference of (*a*) and (*b*), which results in the spectrogram of the transient sounds only. A high ANL can mask transient sounds, as highlighted by the areas surrounded in red. In part, this may also be that some biological signals that blend into mass choruses can also appear as low-amplitude ambient noise. ANL, ambient background noise level; RL, received level. Dark blue, 30 dB re 1 μPa; bright yellow, 90 dB re 1 μPa.

possible to assess it by iteratively finding a minimum cluster number that can explain a user-defined variation threshold [177]. A similar technique may be useful for analysing the diversity of transient sounds.

## 4.4. Source separation and deep learning

Sounds can travel more efficiently under water compared to in air, and low-frequency sounds can often travel long distances. This complicates marine soundscape analyses because it tends to result in the overlap and interference of multiple sounds. Machine learning-based source separation has achieved considerable improvements in speech and music [219,220]. Source separation models can be trained by labelled data or constructed in a purely unsupervised manner [221]. The latter one is also called blind source separation, which requires appropriate assumptions about the behaviour of source signals. Several studies have successfully demonstrated the separation of biological, anthropogenic and abiotic sounds using these methods [128,222]. Further, extensive work has been done by applying hydrophone/microphone array systems to isolate source signals of interest [23,223,224].

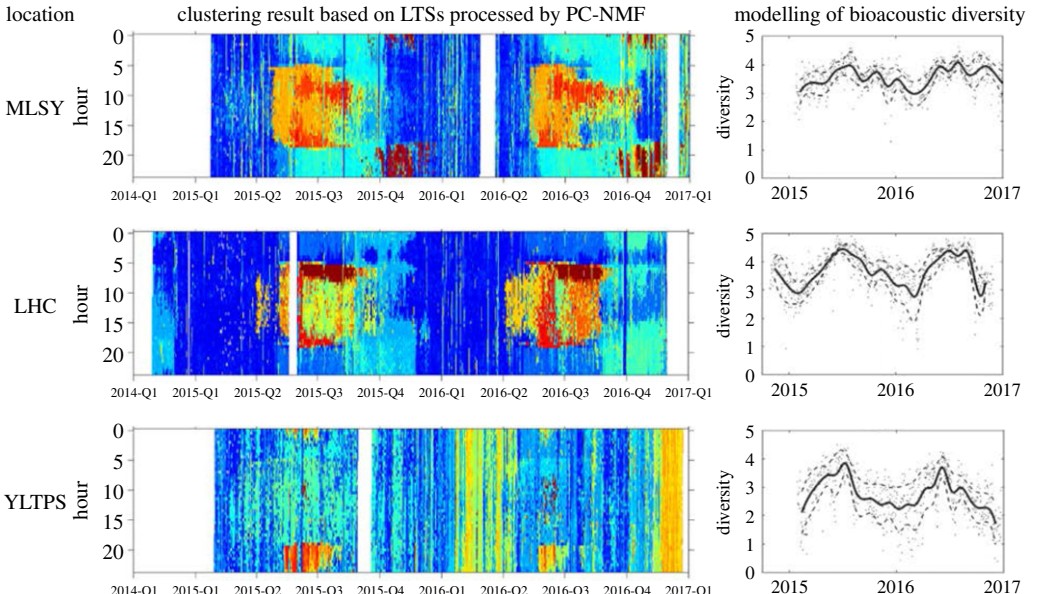

**Figure 5.** Analysis results based on the biological sounds recorded at three observing locations in Taiwan. Within a location, the sounds are separated from the median and difference-based long-term spectrograms. The left panels show the diurnal and seasonal changes in biological sounds by using k-means clustering. Each colour represents a different biological sound with unique spectral features. The right panels show the seasonal change in bioacoustic diversity calculated by measuring the Shannon entropy of the probabilistic distribution of chorus and transient sounds measured via the clustering on the left panels. The solid lines represent the mean diversity, dashed lines represent the standard deviation and dots represent the diversity index measured on each day of recording. The three locations noted are Triangle Mountain (MLSY), Lienhuachih Research Center (LHC) and Taipingshan (hereafter YLTPS), Taiwan. Adapted from [209], see this paper for details.

The recognition of biological sounds can also be improved by applying deep learning techniques. Initial work has been conducted to recognize bird songs, frog calls and marine mammal vocalizations from large acoustic datasets [225–228]. Although deep learning has demonstrated its power in many acoustical applications, it requires prior knowledge of all sound sources of interest and sufficient training data (e.g. labelled acoustic datasets) for each of them, including rare sounds that are relevant in acoustic biodiversity assessment. Unfortunately, such a database is still not available for most marine ecosystems and most soniferous marine animals. On the other hand, unsupervised learning techniques can be employed in separating different sound sources and facilitating the evaluation of acoustic diversity. Sparse coding has been introduced in the extraction of acoustic features, which represent the dictionary of acoustic codes for recorded sound sources [229]. Self-learning algorithms, including non-negative matrix factorization and hidden Markov model, can learn hidden variables and associated temporal weights from audio data [126,230,231]. Source-specific independent subspaces can be derived subsequently by identifying source indicators for hidden variables. This approach has been tested in marine and terrestrial soundscapes, and performed well to separate biological sounds from other sound sources [222,232]. Deep learning methods have also been developed to detect and classify cetacean sounds [233]. While these are not yet being applied in datasets rich with bioacoustic sound types, they do work well despite often high levels of diverse background noise.

## 4.5. Complementary data sources

As acoustic behaviour is often related to environmental and anthropogenic drivers, complementary data sources can provide context to the acoustic patterns observed. For example, salinity, temperature, dissolved oxygen and tidal cycle data can help understand presence, intensity and rate of sound production [97,193,234]. Automatic identification system (AIS), vessel monitoring system (VMS), boat ramp surveys or visual observation can all provide information on vessel activity that masks biological sound or drives a particular change in behaviour [235,236]. Traditional observation techniques can provide a snapshot of physical biodiversity, abundance, species composition, habitat complexity, animal size distribution and surrounding habitat. This is vital validation information [154,184,237] and should, where possible, be ongoing [155] to capture environmental changes that can affect biological sound

production and therefore local soundscape acoustic complexity and diversity. These complementary datasets help validate the bioacoustic diversity at a particular point in time, but also put that time period into context of the wider influences that can change the soundscape on various temporal scales.

# 5. How to listen forward: needs and recommendations for acoustic diversity

While assessing marine acoustic diversity is a nascent field, we have reviewed how prior terrestrial work and other ocean research have provided a foundation; from this foundation, we may build guidelines for designing future acoustic biodiversity studies. We have outlined recommendations below of how to go about this biodiversity assessment task; many of these points highlight the advances or lessons noted above. These suggestions are naturally limited by what we can do now; thus, this section is followed by several needs that may support the field.

(i) *Replication and site-selection*. Within-reef and within-habitat soundscape variation is still being addressed [94,103]; thus, within-site replication is still important to evaluate at a location or habitat. This enables assessments of both α and β biodiversity. Recordings that can be considered replicates should be evaluated to address within-site variation. Because acoustic behaviours are often driven by environmental factors or geophysical conditions, spatio-temporal replicates selected to assess variation between biodiversity or habitat characteristics must have similar geophysical conditions (e.g. salinity, temperature, oxygen cycles, currents tidal cycle). Given that true natural replicates are challenging, sites may be better differentiated by regression-type analyses of gradient differences (i.e. a range of biodiversity or habitat conditions) rather than forced groupings. If we are attempting to evaluate within-site α diversity a nearby outgroup (control site or sites) is essential.

(ii) *Longer timescales*. While some (often visual) surveys may be necessarily short (i.e. limited by SCUBA), an advantage of passive acoustics is that sensors can be deployed for long durations (months, season and beyond). Longer and repeated surveys have a clear and easily applied advantage in reducing observation biases, a vital factor in meeting the clear conservation and management aims of biodiversity assessments. Thus, we recommend sufficient recording time to include and identify the cycles in soniferous behaviours of interest that may be site- and species-specific, rather than short-term or subsampled recording, such as the same period of each day.

(iii) *Evaluate detectability*. No single method or count of animals is perfect, and biodiversity assessments can benefit by correcting animal and species counts by addressing (or estimating) those present, but missed in the assessment. This quantification of *detectability* (the relationship of what is present in a survey area versus what is actually detected) is a major issue in terrestrial biodiversity measurements, and is likely to be similar, or perhaps a greater issue for marine acoustic measures [184], given the substantial differences in things like call levels, spatio-temporal variability, ambient noise levels and propagation noted earlier. The influence of these factors on detectability will vary by habitat (and maybe site) and we often still need to evaluate precisely how much a factor, such as ambient noise, affects call or acoustic diversity detectability. Fortunately, detectability has often been assessed in terrestrial biodiversity assessments [155,238,239]. Indeed, terrestrial detectability assessments have often found similarities to the general trends noted here; particularly because species detection probabilities will rarely approach 1 unless many surveys are conducted [155]. Further, long-term monitoring programmes allow for pooling of data, but also need to correct for variation in detectability when comparing species richness over time and between locations [154]. These parallel lessons suggest that, while many covariates will differ between terrestrial and marine environments, perhaps we can apply the analyses frameworks (such as effective sampling distance, species heterogeneity, parsing of habitat types and by species) to improve the reliability of marine acoustic diversity assessments.

(iv) *Propagation-related aspects* (depth, local bathymetry, substrate) should be identified, and evaluation of weather and other noise sources, especially when comparing among sites. This evaluation supports measurements of detectability.

(v) *Concurrent measurements*. Multiple methods to count animals and species greatly support biodiversity measurements. These are noted above and include visual surveys, measures of rugosity and quantifying physical (currents) and biogeochemical parameters.

(vi) *Pilot data (made available)*. One must try to understand what sources are influencing this variability, or at least the range of variably, so that proper recording guidelines (such as timeframe of observation) are sufficient. Pilot data are valuable in any study, and bioacoustics is no exception. These data should be

made available as electronic supplementary material to guide and support why measurement decisions were made.

(vii) *Reduce noise*. Many acoustic tools exist to improve signal-to-noise analyses, including methods mentioned earlier. Given the propagation challenges of ocean environments and abundance of natural and anthropogenic noise, noise reduction is vital to improving biodiversity analyses and reliability of data. If the noise is limited, one may also remove files with ship or vessel masking noise from analyses. Further, instrument noise of the recording system is a crucial parameter and should be reported.

## 5.1. Bioacoustical analysis

While no clear methods seem successful to extract biodiversity data from acoustic recordings, and applications of even an array of (terrestrial) acoustic diversity indices seem ineffective [39], we see several potential ways forward with acoustic analyses based on initial successes.

This could include identifying, separating and classifying individual signals. Many fish and aquatic invertebrate sounds are found among the low-frequency noise of their habitats; noise reduction related to those signals of interest is key to proper sound detections. The previously discussed source separation techniques offer substantial promise, but need to ensure that sounds with high and lower occurrence rates are both detected and evaluated. Deep learning methods have shown progress in the enhancement of audio signals from a noisy matrix [126,240]. We may also iteratively find acoustic clusters that can explain a user-defined variation threshold [177]. A similar technique may be useful for analysing the diversity of transient sounds. Marine mammal research efforts have long focused on finding signals within noise. Both spectrogram correlation and detection of frequency contour sounds by detecting and grouping spectral peaks should be applied to other sounds [173,241], and at the very least, their detection and classification can be implemented into a more conglomerate acoustic biodiversity metric. Species-specific acoustic neural networks should be tested on detection and classification of multiple species' sounds [233]. Future integration with these techniques in the analysis of marine soundscapes should maximize the potential deliverables regarding the acoustic behaviour of soniferous fishes and invertebrates, vital biological sources important to the variability of marine soundscapes.

More coarsely, without pulling out individual signals, chorus peak detection and area under the curve measurements have proved initially successful [7,242]. This may be a step that can be used when it proves difficult to pull transient and quiet signals from the cacophony of other sounds.

## 5.2. Additional acoustic measurements

Many sounds are inherently directional. The use of hydrophone arrays can help localize sound sources and track individual calling animals [103]. Such data will also allow us to address the volume or area being sampled [243] (figure 6) and assess diversity within this volume and potentially localize individuals (e.g. [244]). Addressing the sampling volume allows us to account for differences in detection ranges. This will allow for better assessments of biological hotspots as well as potentially enumerating the calling animals for density estimations.

Sound directionality may also be determined by measuring acoustic particle motion (velocity or acceleration) [244]. As technology has improved, measurements of particle motion have become increasingly common [245,246] and, as a direct cue for fauna, it is important to test particle motion measurements of the soundscape to investigate its relationship with biodiversity and acoustic communication. In addition, the instrumentation used to measure particle motion offers an alternative method to sound source localization that requires fewer individual sensors (compared with a hydrophone array) and may improve signal detection in noise (e.g. [182,183]). Understanding variations in particle motion, as detected by receptors, facilitates a better understanding of why changes occur. However, in relating the characteristics of a soundscape to the biodiversity within it, whether the animal perceives sound pressure or particle motion is not necessarily important and sound pressure often provides sufficient information to the observer on the sounds produced. Sound pressure is also more readily measurable.

Although acoustic diversity appears to be related to taxonomic diversity, in order to be fully representative of the biodiversity of a habitat or site, better knowledge of the sound sources and their repertoire is necessary, particularly for fish that make a large contribution to marine soundscapes. The traditional method to assess an animal's sound repertoire is to bring it into a tank and record its sounds [247]; however, there is an inherent potential bias from the effects of captivity. The alternative

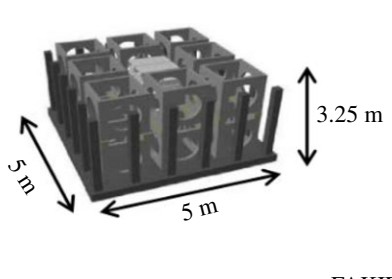
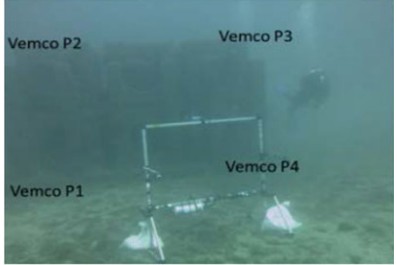

FAKIR

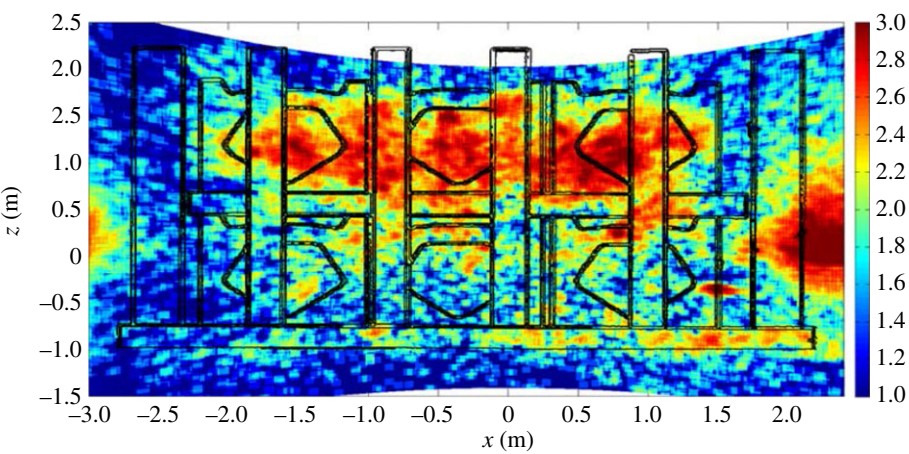

**Figure 6.** Three-dimensional localization of broadband transient sounds produced by invertebrates inhabiting an artificial reef (FAKIR). On the top panel, a schematic and a photographic representation of the artificial reef module. The lower panel represents the acoustic map of localized invertebrate sounds. In red, areas with a higher density (1000 sounds $m^{-2}$ $min^{-1}$), in blue, areas with low densities (10 sounds $m^{-2}$ $min^{-1}$). The upper 'level' of the FAKIR clearly shows biological hotspots. The black lines draw the contours of the artificial reef [243].

is labour-intensive *in situ* validated acoustic observations, such as those taken on re-breather dive equipment [138]. Biologging acoustic-recording tags are increasingly used for marine mammals [248]. A combination of acoustic telemetry tags attached or implanted and three-dimensional passive acoustic localization may alternatively help assessing additional sound source identities, calling rates and propagation distance in the wild.

## 5.3. Open data and open tools

To date, sound archives have been established for most marine mammals and some soniferous fish (e.g. Macaulay Library, Watkins Marine Mammal Sound Database and Moby Sound Archive). However, a comprehensive audio database of all soniferous marine organisms is still not available. A library of labelled fish and invertebrate signals, with metadata such as region recorded, would go a long way in sorting, comparing and analysing large datasets. Additionally, soundscapes in many marine ecosystems, such as the deep ocean, remain relatively unstudied. Under these circumstances, products using acoustic methods for assessing marine biodiversity are limited. In recent years, long-term underwater recordings, such as data obtained by cabled observatories and autonomous recorders (www.iqoe.org/systems), have been archived. These data serve as essential information to investigate the ecosystem-specific soundscape characteristics as well as the relationship between acoustic diversity and marine biodiversity.

Future developments of open data and tools are also critical for expanding the ecological dimensions of acoustic data. An open platform, which allows Internet users to freely access soundscape recordings, will facilitate the identification of biological sounds. A similar approach has been proven to be effective in citizen science (to name a few: iNaturalist, XenoCanto, eBird, Whale FM). The establishment of an open database would not only facilitate public involvement, it could also encourage the future development of analysis toolboxes, while ensuring maintenance of the appropriate control protocols. The Bird Audio

**Table 1.** Open-sourced tools for information retrieval of marine soundscapes.

| toolbox | applications | references |
| --- | --- | --- |
| PAMGuide | describe changes in acoustic heterogeneity | Merchant et al. [204] |
| PAMGuard | detection, classification and localization | Gillespie et al. [249] |
| Triton | visualization of large audio dataset, detection and classification | Wiggins et al. [250] |
| Soundscape Viewer | visualization of large audio datasets, source separation and event clustering | Lin et al. [126] |
| CHORUS | pre-processing and analysis of acoustic data | Gavrilov & Parsons [251] |
| Raven-X | detection of animal vocalizations | Dugan et al. [252] |
| seewave | visualization of audio data, ecoacoustic indices | Sueur et al. [253] |
| soundecology | ecoacoustic indices | Villanueva-Rivera & Pijanowski [254] |
| paPAM | describe sound levels across frequency and time, or impulse characteristics, for acoustic pressure and particle motion | Nedelec et al. [246] |

Detection challenge, for example, has resulted in a growth in the number of deep learning tools applied to avian acoustic research [228].

Utilization of signal processing and audio recognition tools is critical for information retrieval of marine soundscapes. Currently, tools such as PAMGuide, PAMGuard, Triton, Soundscape Viewer and several ecoacoustic packages in the R programming environments have been employed in various acoustic applications of marine biodiversity (table 1). However, it is necessary to develop more tools to meet the needs of different users and project objectives. Open-source tools will be important drivers to promote the future development of advanced techniques in soundscape information retrieval.

Ongoing development of processing techniques can be facilitated through convening an open forum where validated soundscape datasets (i.e. known acoustic signals with matching biodiversity, habitat, geophysical and bathymetric data) are compared by participants in a blind study. Results are then presented and compared in a transparent format to evaluate processing techniques and results, similar to that of GIS mapping and seafloor substrate and habitat characterization (e.g. Shallow Water Survey, Wellington, New Zealand, 2012) or modelling of fish distribution (e.g. Geohab, Lorne, Australia, 2014).

Given the relative ease of collecting acoustic data with modern technology, but also the enormity of many of these datasets (into the terabytes), cloud computing and leveraging powerful processing will probably be necessary to adequately exploit this information. The simplicity of soundscape data collection may also facilitate and improve observations of biodiversity, particularly in places where routine monitoring is limited via other methods. Such monitoring also requires relatively inexpensive acoustic recorders and means to either conduct on-board processing at the recording site or a cost-effective means of relaying relatively large acoustic datasets to the computing cloud. Much of these data collection and analyses should be conveyed to the Web in near-real time.

Building from this capability to link people and data, there is a need for a global field experiment that could be designed to test some of the methods. This would allow us to address how acoustic diversity metric methods vary based on regions and habitat types (to evaluate β-diversity). Ideally, acoustic diversity methods should be generalizable, reproducible and comparable irrespective of geographical and environmental factors. Placing multi-site data or a subset of data, measuring the same parameters, in different parts of the world and in different ecosystems, into common, online repositories would support common analyses to address this need.

# 6. Summary

There is a growing call for soundscape measurements to aid biodiversity assessments and, given the diversity of sounds in aquatic environments, such concepts are promising. Yet, it is vital to also address the limitations of these methods so that they are used properly for management and research purposes. The goal of these efforts is ultimately to improve and broaden biodiversity estimates to better determine and monitor critical conservation areas. Therefore, the tools must ultimately serve the

public and resource managers. Passive acoustics has an advantage over many other methods in that it can be cost-effective to observe over long timescales. In the ocean, sound cues travel efficiently, which is a benefit to detecting sounds but a challenge to mitigate masking lower amplitudes and less frequent sounds, which are critical to conservation monitoring.

We have provided an assessment of this field's progress to date and a critical evaluation of the considerations for ecoacoustic assessments of biodiversity. The bioacoustic environment is immensely variable. One must try to understand what sources are influencing this variability, or at least the range of variability, so that proper recording guidelines (such as timeframe of observation) are followed. These efforts and our recommendations may be particularly valuable when level of biodiversity and community complexity are unclear, and on a sliding (non-binary) scale. Key needed aspects for these acoustical biodiversity assessments include: (i) replication and site-selection, (ii) recording over sufficient timescales, (iii) evaluation of detectability, (iv) evaluation of propagation-related aspects, (v) making concurrent measurements of complementary variables, (vi) collection of pilot data and making those data available, and (vii) seeking to reduce acoustic noise. Further, a suite of established and emerging passive acoustic analysis tools exist, and can potentially be incorporated into developing a metric. We see these tools as having great potential, but we suggest a cautious approach to applying current acoustic indices to assess marine biodiversity. Yet, with key preliminary data, an initial understanding of the patterns and variability of a habitat, and new analytical tools, there is substantial promise in quantifying marine bioacoustic diversity and supporting management needs.

Data accessibility. This article has no additional data.

Authors' contributions. T.A.M. provided the initial draft. All other authors revised and edited this paper equally.

Competing interests. We declare we have no competing interests.

Funding. Funding for development of this article was provided by the collaboration of the Urban Coast Institute (Monmouth University, NJ, USA), the Program for the Human Environment (The Rockefeller University, New York, USA) and the Scientific Committee on Oceanic Research. Partial support was provided to T.A.M. from the National Science Foundation grant OCE-1536782.

Acknowledgements. The IQOE Project, including Jesse Ausubel, Peter Tyack and Bishwajit Chakraborty, brought this team together to address this overall need of examining marine acoustic diversity and supported travel for the team to meet, discuss and build consensus on these issues. We thank them for their initiative and support. We thank Natalie Renier for illustration support.

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
