## [Reviewer comments · Royal Society Open Science]

Review History

RSOS-200149.R0 (Original submission)

Review form: Reviewer 1 (Steve Buckland)

Is the manuscript scientifically sound in its present form?

Yes

Are the interpretations and conclusions justified by the results?

Yes

Is the language acceptable?

Yes

Do you have any ethical concerns with this paper?

No

Have you any concerns about statistical analyses in this paper?

Yes

Recommendation?

Major revision is needed (please make suggestions in comments)

Comments to the Author(s)

I read this paper with interest, but was rather disappointed. The paper provides a wide-ranging review, and gives a useful list of references. However, in places, it came over to me as a 'stream of consciousness', rather unfocussed, and listing a large number of points without really drawing things together. For me, more focus could be achieved by the following.

1. Consider how biodiversity monitoring studies in marine environments should be designed. How do you select sampling locations, how do you sample through time, and for how long do you record at sampling times? At present, you make comments on how it should not be done, but you do not provide recommendations on how it should be done.
2. You indirectly consider the issue of detectability in various places in the paper, but there is no coherent discussion. For example, how might measures of biodiversity be affected during noisy environmental conditions or when there is substantial anthropogenic noise? You note the need to separate different categories of noise, but you do not note that, even if you can separate out the noises from biological sources, data collected when other noise sources are loud will not be comparable with those obtained when other sources are quiet. Detectability is a major issue in terrestrial biodiversity measurement, but it seems it will be a much greater issue for marine acoustic measures, where there are big differences in the noise made by different taxa, and where propagation is so variable.
3. You do not get into details of biodiversity measurement from acoustic data. You note that the acoustic data can be dominated by a single species or group, but you make no recommendations on how to address this issue. You might for example form a measure for marine mammals, another for fish, a third for invertebrates (which of course assumes you can separately measure these components). Trends formed from the different measures might then be combined into a composite index, or might be interpreted as separate indices. This might help reduce the problem of detectability. You might also be able to develop measures of relative detectability, so that biodiversity measures can be weighted to account for differences. You also make comments about the need to monitor transient and rare sounds, but you do not explain why these are an important component of acoustic diversity.
4. Stating clearly what the focus of your paper is might help. For example, are you interested in comparing biodiversity at different locations, or are you interested in quantifying temporal trends at a given location? In the latter case, you might also be interested in how the temporal trend varies by location through a study region. Having a single, clearly-stated focus would allow you to cut out some material, and produce a clearer message, perhaps coupled with a list of recommendations for what needs to be done before acoustic monitoring can achieve your stated objective, or perhaps providing a checklist of issues to consider when setting up a monitoring program, bearing in mind that there may be multiple objectives.

Review form: Reviewer 2

Is the manuscript scientifically sound in its present form?

Yes

Are the interpretations and conclusions justified by the results?

Yes

Is the language acceptable?

Yes

Do you have any ethical concerns with this paper?

No

Have you any concerns about statistical analyses in this paper?

No

Recommendation?

Accept with minor revision (please list in comments)

Comments to the Author(s)

Marine bioacoustics is a quickly expanding field with an ever-increasing and diffuse body of literature. This review provides a thorough summary of many of the current and historical approaches in the field. Overall, the language is clear and easy to read, and the explanations are good. I think the paper is certainly acceptable for publication, but I have a few comments that should be addressed to improve it further.

General Comments

- The review has a major focus on tropical marine systems, often at the expense of other marine or terrestrial habitats, even though many of the approaches are similar or analogous. The problem, though, is that the review goes back and forth between being broadly encompassing of all habitat with acoustics, and then extremely specific focusing on reefs. I'm not sure what the best way is to balance this in the context of a review, but it's worth keeping in mind at the revision stage.
- The review provides a framework for "assessments," but doesn't provide any historical context for where and how biodiversity assessments came about and how/why they are used or implemented. For example, much of the work on acoustic indices is meant to parallel alpha or beta diversity metrics for habitats, but I think it would be useful for the authors to provide some context on how biodiversity assessments came about which can then help inform why acoustic assessments are conducted in their present form; that would also help highlight similarities and differences between the two. This is discussed briefly in lines 69-72, but I think the entire paper would benefit from a bit more detail and context.
- In the discussion of acoustic indices for marine assessments, one critical citation is missing is Buxton et al. 2018 (citation below) that points out that acoustic indices perform far better in terrestrial environments than acoustic ecosystems. This paper is a helpful reminder that while acoustic indices are appealing for a number of reasons, there is some additional validation needed.

Buxton, R. T., M. F. McKenna, M. Clapp, E. Meyer, E. Stabenau, L. M. Angeloni, K. Crooks, G. Wittermyer. 2018. Efficacy of extracting indices from large-scale acoustic recordings to monitor biodiversity. *Conservation Biology* 32:1174-1184.

Specific Comments

- Lines 36-37: worth including high species diversity in the challenges of ocean assessments.
- Line 49: I'd make the strong argument that marine bioacoustics as a field, or even acoustic assessments aren't an "emerging method," as many of these approaches have been around for decades. There are certainly things in this review that are discussed that are new, but the field has been around for a long time.
- Lines 106-109: it would also be worth citing the Simon Link review in applying acoustic diversity to freshwater systems:

Linke, S., T. Gifford, C. Desjonquères, D. Tonolla, T. Aubin, L. Barclay, C. Karaconstantis, M. J. Kennard, F. Rybak, J. Sueur. 2018. Freshwater ecoacoustics as a tool for continuous ecosystem monitoring. *Frontiers in Ecology and the Environment* 16:231-238.

- Lines 116-118: I think it is critical (and missing in the description here) to explain to the readers what some of the fundamental assumptions are about acoustic indices and why they are appropriate for applying to biological sounds, namely that most biological sounds have certain frequency and temporal properties.
- Line 126: The Buxton et al. 2018 and Linke et al. 2018 papers could also be cited here.
- Lines 162-165: If these economic values were published in 2003, they may underestimate the value in current dollars. For example, \$9B in 2003 is approximately ~\$12B-16B in 2018. It might be safer (and less ephemeral) to just say “multibillion” than to provide a discreet valuation.
- Lines 268-269: I think it’s worth describing the recommendations of the Harris et al. 2016 study, namely which indices performed better than others.
- Lines 750-762: This paragraph seems to be more appropriate in the above section B or C, and less-so as part of “source separation.”

Review form: Reviewer 3

Is the manuscript scientifically sound in its present form?

Yes

Are the interpretations and conclusions justified by the results?

No

Is the language acceptable?

Yes

Do you have any ethical concerns with this paper?

No

Have you any concerns about statistical analyses in this paper?

No

Recommendation?

Major revision is needed (please make suggestions in comments)

Comments to the Author(s)

A research framework for assessing marine biodiversity using acoustic methods

This review is meant to be a synthesis of bioacoustic methods available for assessing marine biodiversity borne out of an International Quiet Ocean Experiment biodiversity working group. The review broadly covers available metrics for assessing biodiversity, challenges with applying these metrics to the marine environment, and issues of relevance pertaining to these measurements. The presumed goal of the manuscript is to “improve and broaden biodiversity estimates and tracking to better monitor critical areas” (Lines 884-888). The goal is ambitious and should be lauded; however, this manuscript misses the mark.

The manuscript lacks a cogent and consistent thesis. Instead it broadly reads like the introduction to a dissertation (a slightly biased dissertation, from a student of tropical acoustics). That there is a complete lack of references to polar regions north or south is deeply troubling, as those soundscapes are highly diverse and are perhaps equally or better candidates for acoustic biodiversity monitoring. Similarly, though the authors should be applauded for their knowledge of coral reefs and sonic fish chorusing, they have a bias away from other taxa and regions that is made glaringly obvious by the lack of reference to many sentinel manuscripts. This review does pay lip service to marine mammals, but overall fails to address the role that these highly vocal (and loud) vocalizers play in contributing to global soundscapes both within and beyond the tropics. I encourage the authors to look closely at Antarctic soundscapes (See work by Ilse Van Opzeeland) and Arctic soundscapes (see work by Aaron Thode, Stafford, and Mellinger), to consider the trans latitudinal work by Haver et al, and to consider more broadly the work by Mellinger et al in general.

This work is interesting, and relevant, and addresses many important themes. But is not well structured (i.e. each section does not naturally flow to the next out of a cogent thesis), but instead feels more like a series of building blocks stacked next to one another to generate significant amounts of interesting and important information, but not comprehensively or to any specific end.

If this is meant to be a commentary on the efficacy (or lack thereof) of acoustic diversity indices, please include a broader literature base (e.g. include Buxton et al 2018) and omit some of the latter sections, or alternatively broaden the thesis out to address the current role of soundscape ecology/ecoacoustics. In its current form, this review achieves neither, and isn't comprehensive enough to constitute a review.

Despite this critical review, I do believe this work is interesting and important. I hope the authors will take the time to revise this manuscript as it has the potential to make a lasting impact on the field.

Decision letter (RSOS-200149.R0)

10-Mar-2020

Dear Dr Mooney:

Manuscript ID RSOS-200149 entitled "A research framework for assessing marine biodiversity using acoustic methods" which you submitted to Royal Society Open Science, has been reviewed. The comments from reviewers are included at the bottom of this letter.

In view of the criticisms of the reviewers, the manuscript has been rejected in its current form. However, a new manuscript may be submitted which takes into consideration these comments.

Please note that resubmitting your manuscript does not guarantee eventual acceptance, and that your resubmission will be subject to peer review before a decision is made.

Your resubmitted manuscript should be submitted by 07-Sep-2020. If you are unable to submit by this date please contact the Editorial Office.

on behalf of Professor Len Thomas (Associate Editor) and Kevin Padian (Subject Editor)
openscience@royalsociety.org

Associate Editor Comments to Author (Professor Len Thomas):

Associate Editor: 1

Comments to the Author:

Thank-you for your submission. We have received three independent reviews and, while all three authors find merit in the paper, all comment in various ways on the lack of focus. The manuscript clearly falls short of its aim of being a synthesis of bioacoustic methods available for assessing marine biodiversity. As a working group output, it seems to suffer from being written by committee. Apart from lack of focus, reviewer 1 comments on some statistical oversights, reviewers 2 and 3 on the tropical bias, and reviewer 3 provides a set of detailed comments in an attachment. I am recommending your manuscript be returned for major revision. I hope you decide to make a resubmission addressing the points made by the referees; assuming this is done in a comprehensive way I see no need to send it for re-review, although I reserve the right to do so should I have doubts about your revisions.

Editor comments:

Thanks for your submission. Our AE has distilled the recommendations of the reviewers and recommends a "major revision" decision. I agree with this as it concerns the level of changes needed, for the most part. However our "major decision" timeline is a bit short, and it may take you a while to get the statistical considerations requested by one of the reviewers. A "reject/resub" decision allows that, without prejudice.

I agree with the reviewers that the topics, though relevant, appear somewhat scattered. Sometimes headings that are question-driven rather than topic-driven help to focus the reader's attention on why discussions are relevant or important at a given point. Just a suggestion.

Finally, your review is very methods-oriented, which is fine. However, I'm not sure that readers get much of a sense of exactly how bioacoustic studies improve our understanding of biodiversity. For example, have they detected previously unrecognized species presence or interactions, and to what extent statistically are these assessments increased? For example, we know from studies of culled forests carried out by Krause and his colleagues that an apparently healthy stand of trees that has been logged selectively appears to have not decreased diversity visibly, but the bioacoustic profile of sonograms shows considerable drop in species presence. A few cases would help readers get an idea of how important this field is. Thanks and best wishes for your revision.

Reviewers' Comments to Author:

Reviewer: 1

Comments to the Author(s)

I read this paper with interest, but was rather disappointed. The paper provides a wide-ranging review, and gives a useful list of references. However, in places, it came over to me as a 'stream of consciousness', rather unfocussed, and listing a large number of points without really drawing things together. For me, more focus could be achieved by the following.

1. Consider how biodiversity monitoring studies in marine environments should be designed. How do you select sampling locations, how do you sample through time, and for how long do you record at sampling times? At present, you make comments on how it should not be done, but you do not provide recommendations on how it should be done.
2. You indirectly consider the issue of detectability in various places in the paper, but there is no coherent discussion. For example, how might measures of biodiversity be affected during noisy environmental conditions or when there is substantial anthropogenic noise? You note the need to separate different categories of noise, but you do not note that, even if you can separate out the noises from biological sources, data collected when other noise sources are loud will not be comparable with those obtained when other sources are quiet. Detectability is a major issue in terrestrial biodiversity measurement, but it seems it will be a much greater issue for marine acoustic measures, where there are big differences in the noise made by different taxa, and where propagation is so variable.
3. You do not get into details of biodiversity measurement from acoustic data. You note that the acoustic data can be dominated by a single species or group, but you make no recommendations on how to address this issue. You might for example form a measure for marine mammals, another for fish, a third for invertebrates (which of course assumes you can separately measure these components). Trends formed from the different measures might then be combined into a composite index, or might be interpreted as separate indices. This might help reduce the problem of detectability. You might also be able to develop measures of relative detectability, so that biodiversity measures can be weighted to account for differences. You also make comments about the need to monitor transient and rare sounds, but you do not explain why these are an important component of acoustic diversity.
4. Stating clearly what the focus of your paper is might help. For example, are you interested in comparing biodiversity at different locations, or are you interested in quantifying temporal trends at a given location? In the latter case, you might also be interested in how the temporal trend varies by location through a study region. Having a single, clearly-stated focus would allow you to cut out some material, and produce a clearer message, perhaps coupled with a list of recommendations for what needs to be done before acoustic monitoring can achieve your stated objective, or perhaps providing a checklist of issues to consider when setting up a monitoring program, bearing in mind that there may be multiple objectives.

Reviewer: 2

Comments to the Author(s)

Marine bioacoustics is a quickly expanding field with an ever-increasing and diffuse body of literature. This review provides a thorough summary of many of the current and historical approaches in the field. Overall, the language is clear and easy to read, and the explanations are good. I think the paper is certainly acceptable for publication, but I have a few comments that should be addressed to improve it further.

General Comments

- The review has a major focus on tropical marine systems, often at the expense of other marine or terrestrial habitats, even though many of the approaches are similar or analogous. The problem, though, is that the review goes back and forth between being broadly encompassing of all habitat with acoustics, and then extremely specific focusing on reefs. I'm not sure what the best way is to balance this in the context of a review, but it's worth keeping in mind at the revision stage.
- The review provides a framework for "assessments," but doesn't provide any historical context for where and how biodiversity assessments came about and how/why they are used or implemented. For example, much of the work on acoustic indices is meant to parallel alpha or beta diversity metrics for habitats, but I think it would be useful for the authors to provide some context on how biodiversity assessments came about which can then help inform why acoustic assessments are conducted in their present form; that would also help highlight similarities and differences between the two. This is discussed briefly in lines 69-72, but I think the entire paper would benefit from a bit more detail and context.
- In the discussion of acoustic indices for marine assessments, one critical citation is missing is Buxton et al. 2018 (citation below) that points out that acoustic indices perform far better in terrestrial environments than acoustic ecosystems. This paper is a helpful reminder that while acoustic indices are appealing for a number of reasons, there is some additional validation needed.

Buxton, R. T., M. F. McKenna, M. Clapp, E. Meyer, E. Stabenau, L. M. Angeloni, K. Crooks, G. Wittermyer. 2018. Efficacy of extracting indices from large-scale acoustic recordings to monitor biodiversity. *Conservation Biology* 32:1174-1184.

Specific Comments

- Lines 36-37: worth including high species diversity in the challenges of ocean assessments.
- Line 49: I'd make the strong argument that marine bioacoustics as a field, or even acoustic assessments aren't an "emerging method," as many of these approaches have been around for decades. There are certainly things in this review that are discussed that are new, but the field has been around for a long time.
- Lines 106-109: it would also be worth citing the Simon Link review in applying acoustic diversity to freshwater systems:

Linke, S., T. Gifford, C. Desjonquères, D. Tonolla, T. Aubin, L. Barclay, C. Karaconstantis, M. J. Kennard, F. Rybak, J. Sueur. 2018. Freshwater ecoacoustics as a tool for continuous ecosystem monitoring. *Frontiers in Ecology and the Environment* 16:231-238.

- Lines 116-118: I think it is critical (and missing in the description here) to explain to the readers what some of the fundamental assumptions are about acoustic indices and why they are appropriate for applying to biological sounds, namely that most biological sounds have certain frequency and temporal properties.
- Line 126: The Buxton et al. 2018 and Linke et al. 2018 papers could also be cited here.
- Lines 162-165: If these economic values were published in 2003, they may underestimate the value in current dollars. For example, \$9B in 2003 is approximately ~\$12B-16B in 2018. It might be safer (and less ephemeral) to just say "multibillion" than to provide a discreet valuation.

- Lines 268-269: I think it's worth describing the recommendations of the Harris et al. 2016 study, namely which indices performed better than others.
- Lines 750-762: This paragraph seems to be more appropriate in the above section B or C, and less-so as part of "source separation."

Reviewer: 3

Comments to the Author(s)

A research framework for assessing marine biodiversity using acoustic methods

This review is meant to be a synthesis of bioacoustic methods available for assessing marine biodiversity borne out of an International Quiet Ocean Experiment biodiversity working group. The review broadly covers available metrics for assessing biodiversity, challenges with applying these metrics to the marine environment, and issues of relevance pertaining to these measurements. The presumed goal of the manuscript is to "improve and broaden biodiversity estimates and tracking to better monitor critical areas" (Lines 884-888). The goal is ambitious and should be lauded; however, this manuscript misses the mark.

The manuscript lacks a cogent and consistent thesis. Instead it broadly reads like the introduction to a dissertation (a slightly biased dissertation, from a student of tropical acoustics). That there is a complete lack of references to polar regions north or south is deeply troubling, as those soundscapes are highly diverse and are perhaps equally or better candidates for acoustic biodiversity monitoring. Similarly, though the authors should be applauded for their knowledge of coral reefs and sonic fish chorusing, they have a bias away from other taxa and regions that is made glaringly obvious by the lack of reference to many sentinel manuscripts. This review does pay lip service to marine mammals, but overall fails to address the role that these highly vocal (and loud) vocalizers play in contributing to global soundscapes both within and beyond the tropics. I encourage the authors to look closely at Antarctic soundscapes (See work by Ilse Van Opzeeland) and Arctic soundscapes (see work by Aaron Thode, Stafford, and Mellinger), to consider the trans latitudinal work by Haver et al, and to consider more broadly the work by Mellinger et al in general.

This work is interesting, and relevant, and addresses many important themes. But is not well structured (i.e. each section does not naturally flow to the next out of a cogent thesis), but instead feels more like a series of building blocks stacked next to one another to generate significant amounts of interesting and important information, but not comprehensively or to any specific end.

If this is meant to be a commentary on the efficacy (or lack thereof) of acoustic diversity indices, please include a broader literature base (e.g. include Buxton et al 2018) and omit some of the latter sections, or alternatively broaden the thesis out to address the current role of soundscape ecology/ecoacoustics. In its current form, this review achieves neither, and isn't comprehensive enough to constitute a review.

Despite this critical review, I do believe this work is interesting and important. I hope the authors will take the time to revise this manuscript as it has the potential to make a lasting impact on the field.

Author's Response to Decision Letter for (RSOS-200149.R0)

See Appendix A.

RSOS-201287.R0

Review form: Reviewer 1 (Steve Buckland)

Is the manuscript scientifically sound in its present form?

Yes

Are the interpretations and conclusions justified by the results?

Yes

Is the language acceptable?

Yes

Do you have any ethical concerns with this paper?

No

Have you any concerns about statistical analyses in this paper?

No

Recommendation?

Accept as is

Comments to the Author(s)

Thank you for your positive responses to review comment. This is a useful review.

Decision letter (RSOS-201287.R0)

Dear Dr Mooney,

I am pleased to inform you that your manuscript entitled "A research framework for assessing marine biodiversity using acoustic methods" is now accepted for publication in Royal Society Open Science.

Royal Society Open Science operates under a continuous publication model. Your article will be published as soon as it is ready for publication, and this will be the final version of the paper. As such, it can be cited immediately by other researchers. As the issue version of your paper will be the only version to be published I would advise you to check your proofs thoroughly as changes cannot be made once the paper is published.

Articles are normally press released. For this to be effective we set an embargo on news coverage corresponding to the publication date of the article. We request that news media and the authors do not publish stories ahead of this embargo (when final version of the article is available). Please see the Royal Society Publishing guidance on how you may share your accepted author manuscript at <https://royalsociety.org/journals/ethics-policies/media-embargo/>.

on behalf of Professor Len Thomas (Associate Editor) and Kevin Padian (Subject Editor)
openscience@royalsociety.org

Associate Editor Comments to Author (Professor Len Thomas):

Associate Editor

Comments to the Author:

Thanks for your comprehensive response to the reviewers' and editors' comments. I agree with you that the manuscript is now much improved: it reads better, the issues that were identified are resolved and the scopy broadened.

There are a few typographical issues left to fix, and I ask you to work with the editorial team to resolve these. For example, "i.e." is sometimes used where you clearly mean "e.g."; "complimentarity" where "complementarity" is intended; an "alpha" has an accent on it and the following "beta" is joined to the next word. However I don't want to hold the paper up so am recommending acceptance on the assumption that these (and any other minor typos) will be fixed before publication.

Reviewer comments to Author:

Reviewer: 1

Comments to the Author(s)

Thank you for your positive responses to review comment. This is a useful review.

Appendix A

10-Mar-2020

Dear Dr Mooney:

Manuscript ID RSOS-200149 entitled "A research framework for assessing marine biodiversity using acoustic methods" which you submitted to Royal Society Open Science, has been reviewed. The comments from reviewers are included at the bottom of this letter.

Editor comments:

Thanks for your submission. Our AE has distilled the recommendations of the reviewers and recommends a "major revision" decision. I agree with this as it concerns the level of changes needed, for the most part. However, our "major decision" timeline is a bit short, and it may take you a while to get the statistical considerations requested by one of the reviewers. A "reject/resub" decision allows that, without prejudice.

I agree with the reviewers that the topics, though relevant, appear somewhat scattered. Sometimes headings that are question-driven rather than topic-driven help to focus the reader's attention on why discussions are relevant or important at a given point. Just a suggestion.

Finally, your review is very methods-oriented, which is fine. However, I'm not sure that readers get much of a sense of exactly how bioacoustic studies improve our understanding of biodiversity. For example, have they detected previously unrecognized species presence or interactions, and to what extent statistically are these assessments increased? For example, we know from studies of culled forests carried out by Krause and his colleagues that an apparently healthy stand of trees that has been logged selectively appears to have not decreased diversity visibly, but the bioacoustic profile of sonograms shows considerable drop in species presence. A few cases would help readers get an idea of how important this field is. Thanks and best wishes for your revision.

- Overall: Thank you for this helpful synopsis. We have sought to smooth the manuscript's voice to one (rather than committee), address statistical oversight, and broaden the scope slightly (from tropics/temperate and fishes/inverts) by adding some of the Polar and cetacean work (great suggestions) while noting most of the acoustic diversity work has taken place in temperate and topics. We think this is a better balance. And we addressed all general and detailed comments in an itemize review. The Reviewer's point is followed by a space and a dash (-) and then our comment on how we address each point. Overall, we think these recommendations substantially improved the review. We thank the editors and reviewers for their suggestions and time.
- This point on case-studies is a good point. There certainly examples of identifying previously unrecognized species presence. Data examples regarding how such realizations have affected our understanding of marine community structure seems a bit sparser. However, we added several interesting case studies of acoustics to understand biodiversity and animal presence, including work on marine mammals (right whales, humpbacks), fishes and inverts.
- Additionally, several sections have been edited to not start with questions (good point).

Reviewers' Comments to Author:

Reviewer: 1

Comments to the Author(s)

I read this paper with interest, but was rather disappointed. The paper provides a wide-ranging review, and gives a useful list of references. However, in places, it came over to me as a 'stream of consciousness', rather unfocussed, and listing a large number of points without really drawing things together. For me, more focus could be achieved by the following.

1. Consider how biodiversity monitoring studies in marine environments should be designed. How do you select sampling locations, how do you sample through time, and for how long do you record at sampling times? At present, you make comments on how it should not be done, but you do not provide recommendations on how it should be done.
 - Agreed. This was not fleshed out enough in our 'Future' or 'Summary' section. We have revised the last section substantially. It is now subtitled 'How to listen forward: needs and recommendations for acoustic diversity' outline guidelines for designing future acoustic biodiversity studies including a more detailed acoustic analyses section. We have outlined those recommendations, followed by several needs that may support the field. Yet this not yet a 'how-to' guide. The field is not there yet and we point that out, seeking here to take a more 'view from above' overview, guide the needs but point out the progress and right directions (while steering away from the less helpful paths).
2. You indirectly consider the issue of detectability in various places in the paper, but there is no coherent discussion. For example, how might measures of biodiversity be affected during noisy environmental conditions or when there is substantial anthropogenic noise? You note the need to separate different categories of noise, but you do not note that, even if you can separate out the noises from biological sources, data collected when other noise sources are loud will not be comparable with those obtained when other sources are quiet. Detectability is a major issue in terrestrial biodiversity measurement, but it seems it will be a much greater issue for marine acoustic measures, where there are big differences in the noise made by different taxa, and where propagation is so variable.
 - Good point. We added a small section on detectability, its marine issues, and maybe how to address it in the 'future' section and in the mass and transient sounds sections.
3. You do not get into details of biodiversity measurement from acoustic data. You note that the acoustic data can be dominated by a single species or group, but you make no recommendations on how to address this issue. You might for example form a measure for marine mammals, another for fish, a third for invertebrates (which of course assumes you can separately measure these components). Trends formed from the different measures might then be combined into a composite index, or might be interpreted as separate indices. This might help reduce the problem

of detectability. You might also be able to develop measures of relative detectability, so that biodiversity measures can be weighted to account for differences.

- We sought to do this a bit more in the future section. Note, a composite index has been tried by a few including Buxton et al 2018 and it hasn't paired out. But we tried to get into the 'weeds' a bit more in terms of the acoustic data analyses.

You also make comments about the need to monitor transient and rare sounds, *but you do not explain why these are an important component of acoustic diversity.*

- Thank you. We addressed this point now in the transient sounds section.

4. Stating clearly what the focus of your paper is might help. For example, are you interested in comparing biodiversity at different locations, or are you interested in quantifying temporal trends at a given location? In the latter case, you might also be interested in how the temporal trend varies by location through a study region. Having a single, clearly-stated focus would allow you to cut out some material, and produce a clearer message, *perhaps coupled with a list of recommendations for what needs to be done before acoustic monitoring can achieve your stated objective, or perhaps providing a checklist of issues to consider when setting up a monitoring program, bearing in mind that there may be multiple objectives.*

- Good point. We substantially revised the 'goals' paragraph at the end of the introduction and structured the paper somewhat to better provide the list of recommendations and ways forward.

Reviewer: 2

Comments to the Author(s)

Marine bioacoustics is a quickly expanding field with an ever-increasing and diffuse body of literature. This review provides a thorough summary of many of the current and historical approaches in the field. Overall, the language is clear and easy to read, and the explanations are good. I think the paper is certainly acceptable for publication, but I have a few comments that should be addressed to improve it further.

General Comments

- The review has a major focus on tropical marine systems, often at the expense of other marine or terrestrial habitats, even though many of the approaches are similar or analogous. The problem, though, is that the review goes back and forth between being broadly encompassing of all habitat with acoustics, and then extremely specific focusing on reefs. I'm not sure what the best way is to balance this in the context of a review, but it's worth keeping in mind at the revision stage.
- This is a fair point. Most of the acoustic diversity work has been done on temperate and tropical reefs, but we are trying to be inclusive (and as noted by Rev 3) polar sciences have made substantial advances, particularly in the area of marine mammal acoustic detection and discrimination. Thus, there is some need to refer to polar advances, but maintains focus where

most of work is being carried out (and the biodiversity hotspots). So, we stated this goal and aim in the introduction.

- The review provides a framework for “assessments,” but doesn’t provide any historical context for where and how biodiversity assessments came about and how/why they are used or implemented. For example, much of the work on acoustic indices is meant to parallel alpha or beta diversity metrics for habitats, but I think it would be useful for the authors to provide some context on how biodiversity assessments came about which can then help inform why acoustic assessments are conducted in their present form; that would also help highlight similarities and differences between the two. This is discussed briefly in lines 69-72, but I think the entire paper would benefit from a bit more detail and context.

- This is an interesting comment. In the Intro, we now provide our interpretation of a historical context and how biodiversity assessments came about in several sentences in the intro. However, this subject in itself could be a review. In order to help focus we stayed succinct. Just a few sentences.

- In the discussion of acoustic indices for marine assessments, one critical citation is missing is Buxton et al. 2018 (citation below) that points out that acoustic indices perform far better in terrestrial environments than acoustic ecosystems. This paper is a helpful reminder that while acoustic indices are appealing for a number of reasons, there is some additional validation needed.

Buxton, R. T., M. F. McKenna, M. Clapp, E. Meyer, E. Stabenau, L. M. Angeloni, K. Crooks, G. Wittemyer. 2018. Efficacy of extracting indices from large-scale acoustic recordings to monitor biodiversity. *Conservation Biology* 32:1174-1184.

- Agreed, we’ve added this paper and contextualized its information in several places. Several other studies have similar suggestions and thus we try to reinforce this idea in the abstract, intro and discussion.

Specific Comments

- Lines 36-37: worth including high species diversity in the challenges of ocean assessments.

- Added. Thanks.

- Line 49: I’d make the strong argument that marine bioacoustics as a field, or even acoustic assessments aren’t an “emerging method,” as many of these approaches have been around for decades. There are certainly things in this review that are discussed that are new, but the field has been around for a long time.

- Added and revised slightly.

- Lines 106-109: it would also be worth citing the Simon Link review in applying acoustic diversity to freshwater systems:

Linke, S., T. Gifford, C. Desjonquères, D. Tonolla, T. Aubin, L. Barclay, C. Karaconstantis, M. J. Kennard, F. Rybak, J. Sueur. 2018. Freshwater ecoacoustics as a tool for continuous ecosystem monitoring. *Frontiers in Ecology and the Environment* 16:231-238.

- Added this ref.

- Lines 116-118: I think it is critical (and missing in the description here) to explain to the readers what some of the fundamental assumptions are about acoustic indices and why they are appropriate for applying to biological sounds, namely that most biological sounds have certain frequency and temporal properties.

- Thank you. Added a short paragraph on this.

- Line 126: The Buxton et al. 2018 and Linke et al. 2018 papers could also be cited here.

- They are added. Thank you.

- Lines 162-165: If these economic values were published in 2003, they may underestimate the value in current dollars. For example, \$9B in 2003 is approximately ~\$12B-16B in 2018. It might be safer (and less ephemeral) to just say “multibillion” than to provide a discreet valuation.

- Revised.

- Lines 268-269: I think it’s worth describing the recommendations of the Harris et al. 2016 study, namely which indices performed better than others.

- We are cautious here. As only one study shows these particular results it would be somewhat inconclusive to make broad recommendation/conclusions. But we discuss the use and outcomes of this and other studies in the following paragraph.

- Lines 750-762: This paragraph seems to be more appropriate in the above section B or C, and less-so as part of “source separation.”

- We have edited this area heavily.

Reviewer: 3

Comments to the Author(s)

A research framework for assessing marine biodiversity using acoustic methods

This review is meant to be a synthesis of bioacoustic methods available for assessing marine biodiversity borne out of an International Quiet Ocean Experiment biodiversity working group. The review broadly covers available metrics for assessing biodiversity, challenges with applying these metrics to the marine environment, and issues of relevance pertaining to these measurements. The presumed goal of the manuscript is to “improve and broaden biodiversity estimates and tracking to better monitor critical areas” (Lines 884-888). The goal is ambitious and should be lauded; however, this manuscript misses the mark.

The manuscript lacks a cogent and consistent thesis. Instead it broadly reads like the introduction to a dissertation (a slightly biased dissertation, from a student of tropical acoustics). That there is a complete lack of references to polar regions north or south is deeply troubling, as those soundscapes are highly diverse and are perhaps equally or better candidates for acoustic biodiversity monitoring.

- Good points. We sought to clarify the goals in the Abs, Intro and Conclusion and added polar references and discussions of those regions including a several sentences on how we balance the variation in work carried out in different regions.

Similarly, though the authors should be applauded for their knowledge of coral reefs and sonic fish chorusing, they have a bias away from other taxa and regions that is made glaringly obvious by the lack of reference to many sentinel manuscripts. This review does pay lip service to marine mammals, but overall fails to address the role that these highly vocal (and loud) vocalizers play in contributing to global soundscapes both within and beyond the tropics. I encourage the authors to look closely at Antarctic soundscapes (See work by Ilse Van Opzeeland) and Arctic soundscapes (see work by Aaron Thode, Stafford, and Mellinger), to consider the trans latitudinal work by Haver et al, and to consider more broadly the work by Mellinger et al in general.

- This is a good point. We added many of these key references and discussed their advances. This is tricky however. For example, Haver et al. (trans latitudinal) are not really suitable for what we are discussing. For starters the Haver article only looks at data up to 2 kHz. One of the challenges with marine mammal soundscapes is that they often tend to work on much, much larger scales, and thus inherently miss the lower amplitude fish sounds unless they are in large choruses. But agreed, we should have acknowledged these other works along the way. We've adjusted the text to reflect and acknowledge the work described.

This work is interesting, and relevant, and addresses many important themes. But is not well structured (i.e. each section does not naturally flow to the next out of a cogent thesis), but instead feels more like a series of building blocks stacked next to one another to generate significant amounts of interesting and important information, but not comprehensively or to any specific end.

- Thank you for this guidance. We sought to make a more coherent thesis at the start (Abs, Intro) and a more thorough set of guidance/conclusions to help get to that specific end.

If this is meant to be a commentary on the efficacy (or lack thereof) of acoustic diversity indices, please include a broader literature base (e.g. include Buxton et al 2018) and omit some of the latter sections, or alternatively broaden the thesis out to address the current role of soundscape ecology/ecoacoustics. In its current form, this review achieves neither, and isn't comprehensive enough to constitute a review.

Despite this critical review, I do believe this work is interesting and important. I hope the authors will take the time to revise this manuscript as it has the potential to make a lasting impact on the field.

- Agreed, added. We had missed that paper. Thank you.

Title: This title does not encompass the manuscript as this manuscript focuses narrowly on tropical and at times temperate oceans only

- I have not addressed this yet as most of the work for acoustic diversity has focused on temperate and tropical seas....

Abstract: Similar to the manuscript, the abstract lacks a thesis and a fulfilled goal.

- We revised the manuscript to provide a clearer thesis and define how we fulfilled that goal.

Introduction:

116-129: This section oversells indices without placing them in the proper broader ecological context. The authors need to define biodiversity and the value of it before lauding indices (moreover, these indices have faced significant criticism which should be introduced much sooner).

- Agreed, we added wording to this affect. "Their success has been variable, and in many cases might serve to be verified by independent groups, but it seems that a combinations of acoustic indices (rather than a single metric) are more effective at predicting bioacoustic activity, Shannon diversity, richness (Buxton et al. 2018)." To be clear (and we understand if there is was confusion) we are not lauding indices and agree with the statement below. Given the statement here and below we are revising accordingly.

129-134: The authors fail to cite Buxton et al, which clearly shows that acoustic indices are suboptimal for the marine environment. Which is generally, the experience real-world researchers face when using them (but it is difficult to publish null results, so many go down the rabbit hole only to have wasted their time without a method of sharing the effort).

- Agreed. Cited and included. We are also getting to this point. But it's challenging as some papers say the opposite so we have to work through it a bit (and the reasons for sometimes ill-suited success).

154-170: This seems very specific to the tropics. What about the significant acoustic biodiversity found at the poles?

- So using acoustics to measure biodiversity and acoustic (bio)diversity are very different things. We are addressing the former. We have edited this paragraph to reflect that we our tropical and temperate focus, but acknowledging some of the key work on polar regions. It now states: "To date, much of the work seeking to leverage acoustics to measure marine biodiversity has taken place in areas of concern or biodiversity "hotspots", areas of high-biodiversity or with high-rates of endemism. This has often been temperate and tropical sites thus this review focuses on these studies. Yet, polar regions are areas were acoustic diversity have been noted (Neumann 2017; Kindermann et al. 2008; Roca and Van Opzeeland 2019), perhaps to a more limited extent, and are certainly regions of interest when monitoring for changes in biodiversity and acoustic diversity (Van Opzeeland et al. 2010; Van Opzeeland 2010).
- We also now note this in a new 'goals' paragraph.

Early Work:

A. Early work on soundscapes encompassed far more than just coral reefs.

- Agreed but we are not really focusing on soundscapes, rather passive acoustics to measure biodiversity (or acoustic diversity). Note, the first sentence of the section “applying acoustic indices to measures of biodiversity”. And the two refs in the first paragraph not about coral reefs... but we have edited and revised this this section a bit to acknowledge some additional work from other environments.

B. Why are marine mammals downplayed as contributors to ocean sound? Particularly in tropical regions where chorusing whales significantly contribute to the winter soundscape. Again, see deep omission of polar regions

- Good point. Added this.

C. Buxton’s work deserves considerable mention here- also why is there no mention of instrument noise? This could severely influence measurements.

- Agreed, we added Buxton’s work twice in this section (but note: her key work from 2018 does not address much in the way of aquatic sounds and metrics besides saying they were not very effective, and probably, because as noted above, it’s hard to publish ‘negative’ data).

III-Considerations

This seems like it should be the highlight of the paper, and all else in service of this section, however, it falls short on offering tangible objectives or clear paths forward.

- We have revised this section substantially and outline clear paths forward in section V.

376-385: Why no mention of instrument noise?

- We’re a little confused by what you mean here but if you mean the noise floor of the instrument, agreed, that’s important. There’s no getting below that (it can clearly affect sound detection) but it also doesn’t belong in this section. We added that to ‘Listening forward’ Section V.

401-413: Marine mammal get some lip service here, but what about seals (a huge biological contributor to ocean sound in temperate and polar oceans)- similarly blue, fin, and humpback whales are large contributors.

- Agreed, this was a bit fish-centric. Adjusted. Also, in the section “B Challenges posed by variability” we added more about the dominance of marine mammal calls including papers on a range of species.

419- Should read “some” fish choruses- some choruses are quiet

- Corrected.

438-453: what about ice noise? What about icequakes what about seasonal variation

associated with migratory animals?

- Agreed added seasonal variation and ice to the geophony above. Keep in mind, there are lots of 'noise' sources and this is not a review on types of noise. We simply cannot cite everything, and we appreciated the Reviewer's patience.

481-484: another area where marine mammals are too broadly generalized (humpback song is not a transient signal on tropical breeding grounds- see Haver et al 2019) and polar systems are completely omitted

- This is confusing. Haver et al show that humpback song is seasonal on breeding grounds. Yes, we agree that it also occurs on feeding grounds. This was not the first reference to show this. Also, we must keep the focus of this study, acoustics to measure biodiversity and have to focus on where that has been done...

487-495: This seems like a very cursory intro, it should either be expanded or removed

- This section on transient sound has been revised to be a bit more detailed. The first paragraph now reads "Transient sounds are referred to here as all uncommon biological sounds that occur in low abundances or generally do not produce mass phenomena but, nonetheless, are major important components in acoustic diversity assessments. Indeed, rare observations, acoustic or otherwise, greatly limit probability of detection and thus raw counts of individuals will be biased toward more gregarious, larger, louder or easily detected species leading toward erroneous richness measurements (Anderson et al. 2015; Meyer et al. 2011). Further, detecting a high number of rare species is typical of many surveys. Accurately capturing this number is crucial as rarity of species and overall site richness has implications for threat status and extinction risk, and the number of rare species is also used to establish spatial conservation priorities (Villalobos et al. 2013; Colwell 2009)."

502: Be more specific, which "level"

- Amplitude. Revised.

504: What about within species variation?

- Added a respective sentence to later in the paragraph.

511-515: Need much more to explain this.

- Expanded the section on propagation.